# Unlocking the secrets of SARS-CoV-2 nsp3 by combining experiments with AlphaFold2 domain prediction

Maximilian Edich[1],* , Yunyun Gao[1],* , David C Briggs[2], Andrea Thorn[1,3]

**Nonstructural protein 3 (nsp3) is crucial for SARS-CoV-2 infection. It is the largest protein of the virus with roughly 2000 residues, and a major drug target. However, because of its size, disordered regions, and transmembrane domains, the atomic structure of the whole protein has not yet been established. Only 10 out of its 16 domains were individually determined in experiments. Here, we demonstrate how structural bioinformatics, AI-based fold prediction, and traditional experiments complement each other and can shed light on the makeup of this important protein, both in SARS-CoV-2 and in related viruses. Our method can be generalized for other multidomain proteins. Our prediction-based approach reveals a previously undescribed folded domain, which we could confirm experimentally. Our research also suggests a potential function of the domain Y1: this domain may be involved in the assembly of nsp3, nsp4, and nsp6 into the hexameric pore, which was discovered by electron tomography and exports RNA into the cytosol. The Y1 hexamer, however, could not be expressed on its own. We revise domain segmentation and nomenclature of nsp3 domains.**

## Introduction

### Coronavirus biology

Coronavirus genomes encode several nonstructural proteins (nsps), which permit the virus to deregulate the host immune system and to reproduce within the host cell (1). Upon infection, the nonstructural proteins are translated together as the large polyproteins pp1a and pp1ab, where the latter one is translated as a consequence of a ribosomal frameshift (2). The polyprotein pp1a consists of 10 nsps, whereas pp1ab comprises 16 proteins. The polyproteins are cleaved into functional nonstructural proteins by the viral papain-like proteases encoded in nsp3 and the main protease (nsp5) (3). Depending on the species, nsp3 has one or two papain-like protease domains (4). Without these proteases, the

infection cycle cannot progress, making nsp3 an important drug target in the fight against COVID-19. In this work, we focus on the murine hepatitis virus strain A59 (MHV) and the two sarbecoviruses SARS-CoV-1 and SARS-CoV-2, which are of great interest because of the COVID-19 pandemic. All three belong to the genus *Betacoronavirus*, which is together with *Alphacoronavirus*, *Gammacoronavirus*, and *Deltacoronavirus* part of the subfamily *Orthocoronavirinae*. Those belong to the family of *Coronaviridae*, which in turn belongs to the order *Nidovirales* and finally to the positive single-stranded RNA viruses.

Nonstructural protein 3 (nsp3) accounts for 20% of the viral RNA, with 2,006 amino acid residues in MHV, 1922 in SARS-CoV-1, and 1945 in SARS-CoV-2, respectively. Depending on the species, these residues form up to 16 domains (Fig 1), including two transmembrane helices, which attach the protein to the membrane of double-membrane vesicles (DMVs) (5) inside the host cell. These vesicles originate from the endoplasmic reticulum during infection and house the replication of the viral genome (5). Although most regions of nsp3 are on the cytoplasmic side, one short region extends into the vesicle's lumen (Fig 1A) and interacts there with the lumenal domain of nsp4 to mediate double-membrane vesicle formation (6). Along with nsp4 and nsp6, nsp3 assembles into large complexes with a sixfold symmetry, which function as pores, exporting the newly replicated viral RNA into the cytosol (7). Subsequently, the RNA interacts with viral nucleocapsid proteins and forms ribonucleoprotein complexes, which assemble with the other structural proteins into new virions, ready for the next infection (8).

### Nonstructural protein 3 domains

The N-terminal domain of nsp3 is known as **ubiquitin-like domain 1 (Ubl1)** and interacts with the nucleocapsid protein (10, 11). It is followed by a long, disordered domain connecting Ubl1 with the rest of nsp3, the **hypervariable region** (9, 12). The next domain of nsp3 in MHV is the papain-like protease **PL^pro**, performing the cleavage of nsp1 from the polyproteins, whereas in *Sarbecovirus*, this domain is absent (4). **Macrodomain 1 (Mac1)**, also known as

[1]Institut für Nanostruktur und Festkörperphysik, Universität Hamburg, Hamburg, Germany    [2]Signalling and Structural Biology Laboratory, Francis Crick Institute, London, UK    [3]Department AI and Biomolecular Structures, Helmholtz-Zentrum Berlin, Berlin, Germany

Correspondence: andrea.thorn@helmholtz-berlin.de
*Maximilian Edich and Yunyun Gao contributed equally to this work

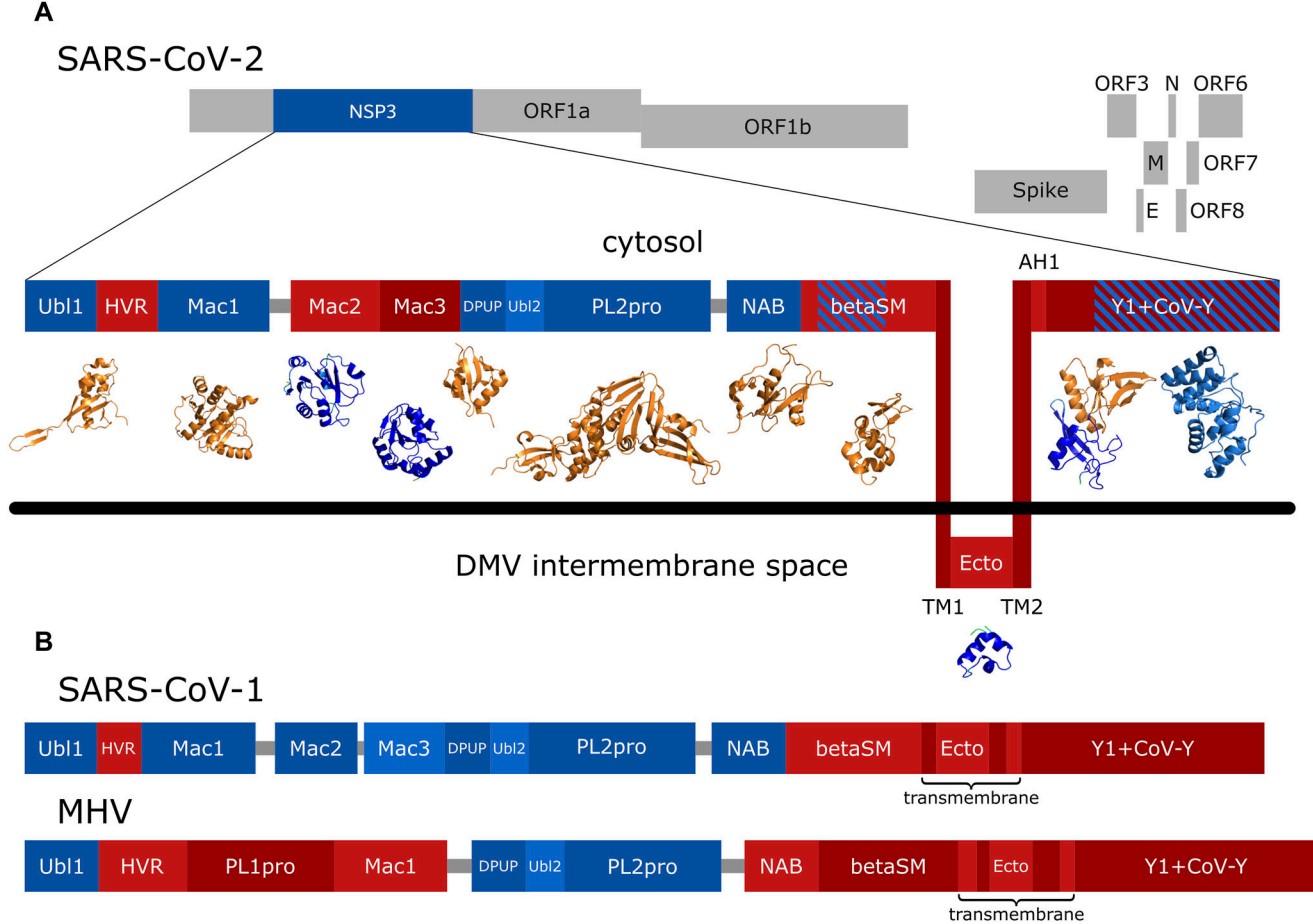

**Figure 1. Domain overview of nsp3 from both sarbecoviruses and MHV.**
**(A)** Position and size of nsp3 on the polyprotein, as well as all domains (blue/red boxes) and larger linkers (gray lines). Blue domains are associated with experimentally solved folding structures in the PDB; red domains are not experimentally solved or are intrinsically disordered, such as HVR (9); red domains with blue stripes are solved partially. Depicted 3D structures in orange are from the PDB (see Table 1 for PDB codes), whereas the blue structures are predicted by AlphaFold2. Membrane topology of SARS-CoV-2 is based on the results of reference 5. **(B)** Domains of SARS-CoV-1 and MHV. Depicted domain ranges are mentioned in the results as preliminary ranges and are listed in Table S1. Domain ranges of SARS-CoV-1 are according to reference 4; ranges of SARS-CoV-2 and MHV are based on sequence alignments with those of SARS-CoV-1 and experimental structures.

ADP-ribose phosphatase, was shown to reverse the human PARP14-derived ADP-ribosylation (13), a mechanism involved in antiviral defense, making Mac1 another well-researched nsp3 drug target (14). Macrodomain 1 is followed by a linker and a region preceding PL2$^{pro}$. For *Sarbecovirus* nsp3, this region was previously known as the SARS-unique domain, consisting of **macrodomain 2 (Mac2)**, **macrodomain 3 (Mac3),** and the **domain preceding Ubl2 and PL2$^{pro}$ (DPUP)**. However, only Mac2 and Mac3 are unique to *Sarbecovirus*, whereas MHV possesses a DPUP-like region (15). Next comes the **ubiquitin-like domain 2 (Ubl2)**, which is seen as a subdomain of **PL2$^{pro}$**. This (in MHV second) papain-like protease is an essential enzymatic domain and therefore a drug target (4, 16). It cleaves the polyproteins between nsp2 and nsp3, between nsp3 and nsp4, and in *Sarbecovirus* in addition between nsp1 and nsp2 (4). PL2$^{pro}$ is separated from the consecutive **nucleic acid–binding domain (NAB)** by a ~30 residue linker. However, as we will show below, this "linker" is a folded and conserved domain specific to *Betacoronavirus*. The last third of nsp3 consists of a

betacoronavirus-specific marker domain (*β*SM), the transmembrane region with a lumenal domain (known as **ectodomain**), and the C-terminal region (4). This C-terminal region consists of the **nidovirus-conserved domain of unknown function (Y1)** and **CoV-Y** (17), whereas two published structures (PDB codes 7RQG (18); 8F2E (19)) and a recent paper (19) suggest to divide both domains into two subdomains.

Currently, experimentally determined structures are only available for the domains Ubl1, Mac1, Mac2, Mac3, DPUP, Ubl2, PL2$^{pro}$, NAB, part of *β*SM, the second half of Y1, and CoV-Y (see Fig 1). Most of the available PDB depositions feature either Mac1 or PL2$^{pro}$, as pointed out by the Coronavirus Structural Task Force (20, 21). The atomic structures of the transmembrane region and part of the C-terminal region were not experimentally elucidated, although many useful insights have been made using nuclear magnetic resonance (NMR) spectroscopy (9, 10, 11, 12, 19). However, novel structure prediction methods enable a first glimpse into their potential folds.

### AI-based fold prediction

In August 2021, the structure prediction software AlphaFold2 (22) became publicly available and enabled a first look into the 3D structure of the undetermined nsp3 domains. AlphaFold2 is based on convolutional neural networks with a two-tower transformer architecture, using coevolution of sequentially remote but spatially close amino acids. The networks were trained on the PDB and multisequence alignments of protein sequences (23). AlphaFold2 predictions closely resemble experimentally determined structures in terms of backbone fold, especially in the case of small and stably folded proteins (24, 25), and come with a confidence score for each residue. Although not intended, this confidence metric, known as the predicted local distance difference test (pLDDT), can be an indicator of disorder (26 *Preprint*, 27 *Preprint*, 28). Another metric, the predicted alignment error (pAE), assists in the recognition and evaluation of subdomains and their relative alignment (29).

In our work, we used AlphaFold2 to predict the structures of all regions of nsp3 from MHV, SARS-CoV-1, and SARS-CoV-2, which were used along experimentally determined structures and the provided metrics from AlphaFold2 to determine domain ranges. This led to the discovery of a new domain and the hexameric assembly of the nsp3 C terminus, followed by experimental validation. In the following, we will demonstrate (A) how we used AlphaFold2 iteratively for construct design and for definition of domain ranges; (B) investigation and validation of the linker domain; and finally (C) analysis of the predicted Y1+CoV-Y hexamer.

# Results

Nonstructural protein 3 (nsp3) is a large multidomain protein present in all coronaviruses. Together with other proteins, it assembles into a hexameric complex that exports new copies of the viral genome into the cytosol and plays therefore an essential role in the viral replication cycle.

We unified the domain boundaries of the individual nsp3 domains, as these were based on few experimental data and were contradictory between publications. During this process, we developed a method for the detection of domains in a multidomain protein based on features of AlphaFold2. With this, we discovered an undescribed domain and validated its fold via a SAXS (small-angle X-ray scattering) experiment. Investigation of the C-terminal Y1 domain led to the hypothesis of Y1 being a major driving force in the assembly of the hexameric RNA-exporting channel. This hypothesis is supported by our bioinformatics analysis, but requires further experimental assessment.

### Using AlphaFold2 for domain boundary determination and construct design

#### Preliminary domain ranges
The number of domains and the exact domain boundaries for nsp3 of SARS-CoV-1 and the related murine hepatitis virus (MHV) changed a lot over the time (17). This led to contradictory

information in the current literature and to partially incomplete gene annotations, especially in the case of SARS-CoV-2. Since the definition of the last domain ranges (4), numerous experimentally determined structures of nsp3 domains emerged, which enable a more complete and accurate domain definition. Furthermore, structure prediction gives insight into currently unresolved regions of nsp3. Here, we aimed at assigning domain ranges to all nsp3 domains of three distinct viruses and unified existing naming conventions. To accomplish this, we used recent experimental structures and AI-based fold prediction.

The available experimental structures cover only a fraction of nsp3. In order to augment domain ranges via structure prediction and investigate unresolved domains, input sequences for Alpha-Fold2 (22) had to be designed. These input sequences were based on sequences from experimentally determined structures and on the domain ranges listed in reference 4 for SARS-CoV-1 and MHV, as well as on the genome annotation entry YP_009742610.1 from NCBI for SARS-CoV-2. Additional information was provided by trans-membrane domain predictions via TMHMM 2.0 (30). Together, this was sufficient to define preliminary domain ranges, which cover all ~2,000 residues of nsp3.

Because the number of experimentally solved domains was limited for SARS-CoV-2 and MHV, we first defined the preliminary domain ranges for SARS-CoV-1. The list was then completed for the other two viruses via sequence alignments against each SARS-CoV-1 domain (see Table S1 for all preliminary domain ranges).

#### Sequence alignments of preliminary domain ranges
To estimate the similarity of domains between both sarbecoviruses and MHV, sequences of the preliminary ranges were compared in local pairwise sequence alignments (see Table S2). Sequence identities were mostly high between the two sarbecoviruses and lower between sarbecoviruses and MHV. From the 18 domains and large linkers, four regions are assumed to be intrinsically disordered (4). From those, however, only three (linker Mac1-Mac2, hypervariable region, and betacoronavirus-specific marker domain) show relatively low identities (41–69% between sarbecoviruses). With 80%, the sequence identity of the linker between PL2[pro] and NAB is comparable to the ordered regions with identities of 71–90%. The two C-terminal domains Y1 and CoV-Y are among the domains with the highest identity (88.1% for Y1 and 90.2% for CoV-Y between sarbecoviruses), which are studied together with the PL2[pro]-NAB linker in the following sections.

#### Comparison between experimentally known structures and AlphaFold2 predictions
The preliminary domain ranges (see Table S1) were used to generate input sequences for AlphaFold2 (22), covering the entire nsp3 residue range. If an experimentally determined structure of a domain was available, the AlphaFold2 prediction was compared with it (Table 1 lists the respective root mean square deviations [RMSD]).

Most cases show RMSD values below 1 Å and high per-residue confidence (high predicted local distance difference test [pLDDT] values) across the whole structure. The two exceptions, Ubl1 domains from SARS-CoV-1 and MHV, contain a flexible N terminus (9, 32), which had pLDDT values below 50. Although small

**Table 1. Comparison of AlphaFold2 predictions to corresponding experimental structures with RMSD values.**

| Name | Abbreviation | SARS-CoV-2 | SARS-CoV-1 | MHV |
|---|---|---|---|---|
| Ubiquitin-like domain 1 | Ubl1 | 0.5 Å (7KAG) | 1.3 Å (2GRI) | 2.7 Å (2M0A) |
| Macrodomain 1 | Mac1 | 0.3 Å (7KQP) | 0.3 Å (2ACF) | — |
| Macrodomain 2 | Mac2 | — | 0.4 Å (6YXJ) | — |
| Macrodomain 3 | Mac3 | — | 0.7 Å (2JZD) | — |
| Domain preceding Ubl2 and PL2pro | DPUP | 0.3 Å (7THH) | 0.6 Å (2KAF) | 0.4 Å (4YPT) |
| Ubiquitin-like domain 2 | Ubl2 | 0.2 Å (7D6H) | 0.2 Å (2FE8) | 0.3 Å (5WFI) |
| Papain-like protease 2 | PL2pro | 0.7 Å (7D6H) | 0.5 Å (5TL7) | 0.5 Å (5WFI) |
| Nucleic acid–binding domain | NAB | 0.4 Å (7LGO) | 0.8 Å (2K87) | — |
| Betacoronavirus-specific marker domain | ßSM | 0.9 Å (7T9W) | — | — |
| Nidovirus-conserved domain of unknown function subdomain b | Y1b | 0.4 Å (8F2E) | — | — |
| Coronavirus-specific C-terminal domain | CoV-Ya | 0.4 Å (8F2E) | — | — |
| Coronavirus-specific C-terminal domain | CoV-Yb | 0.4 Å (7RQG) | — | — |

We used the rank 1 predictions from AlphaFold2 and calculated the RMSD with PyMOL (31). In case of Mac1 and PL2pro for SARS-CoV-2, where a great number of published structures exists, we compared to the structures with the highest resolution and picked the one with the lowest RMSD. For all other domains, we tested all PDB entries and listed here the lowest RMSD, with the tested PDB structure behind the RMSD value. In case of 8F2E, atoms were removed before alignment to match the exact subdomains.

misalignments were observed for loops and secondary structure elements, the number and presence of such elements were correct in all cases listed in Table 1.

### Classification into regions of order and disorder

AlphaFold2 predictions come with a confidence score (pLDDT, predicted local distance difference test) and a predicted alignment error (pAE) for each residue, which evaluate the local and global folding, respectively. We combined pLDDT and the predicted 3D fold to differentiate between regions of order and disorder. Individual domains within one large segment of order are distinguished via the pAE. Ordered folds are characterized by multiple secondary structure elements and pLDDT values above 80. Loops that are either surrounded by such regions or show high pLDDT values are also considered ordered. Potentially disordered regions on the other hand are lacking secondary structure elements and have pLDDT values below 50. Fig 2 shows our decision tree based on these criteria, used to classify each region into potentially ordered or disordered.

Another property for differentiation is the orientation of the termini. Aligning multiple fold predictions leads to a near-perfect overlap of ordered sections, whereas disordered termini point away from the fold in random orientations. Fig 3A illustrates this phenomenon well on predictions of the domain Ubl1 with its disordered N terminus. The predicted, stretched-out "barbed-wire" conformations are furthermore inconsistent with normal main-chain torsions, as described by reference 28.

A last special case covers ordered regions, which are predicted with well-defined secondary structure elements but with low pLDDT. The prediction of the βSM domain is a good example (Fig 3B). The input sequence covers a large disordered N terminus, an ordered central part, and a large disordered C terminus. A low pLDDT is assigned to the termini, thus correctly implying intrinsic

disorder for that region. However, these termini influence the confidence of the central fold negatively. Cropping these termini and resubmitting the sequence of just the ordered part lead to a high-confidence prediction (Fig 3C). Furthermore, the new prediction is more accurate with the RMSD to the experimentally solved structure (PDB: 7T9W) dropping from 0.9 to 0.5 Å. The C-terminal helices keep low pLDDT values even after sequence cropping, which classifies it as potentially disordered.

### Final determination of domain boundaries

The iterative method of predicting, classifying, cropping, and resubmitting for prediction was used to identify all regions of order and disorder within nsp3. The boundaries between ordered and disordered regions set the final domain ranges, and in case of large segments of order, the predicted alignment error (pAE) was used for domain separation. The resulting pAE matrices are contained in Fig S1.

One large region of order comprised the domains Mac3, DPUP, Ubl2, PL2pro, a newly discovered domain, and NAB. Because it was not possible to distinguish between Ubl2, PL2pro, and the new fold following PL2pro (from here named "linker domain") in the pAE matrix, ranges from literature and experimentally determined structures were used to define domain boundaries. The newly discovered linker domain is further examined in Section B.

The final domain ranges are listed in Table 2. Noteworthy discoveries are a folded helix in the otherwise disordered hyper-variable region for MHV (nsp3 residues 230–241); the aforementioned linker domain between PL2pro and NAB; a fold in the ectodomain; and a full prediction of Y1. None of these have been experimentally observed before. All regions of potential disorder in nsp3 make up 357 residues (18.4%) in SARS-CoV-2, 344 (17.9%) in SARS-CoV-1, and 506 (25.2%) in MHV.

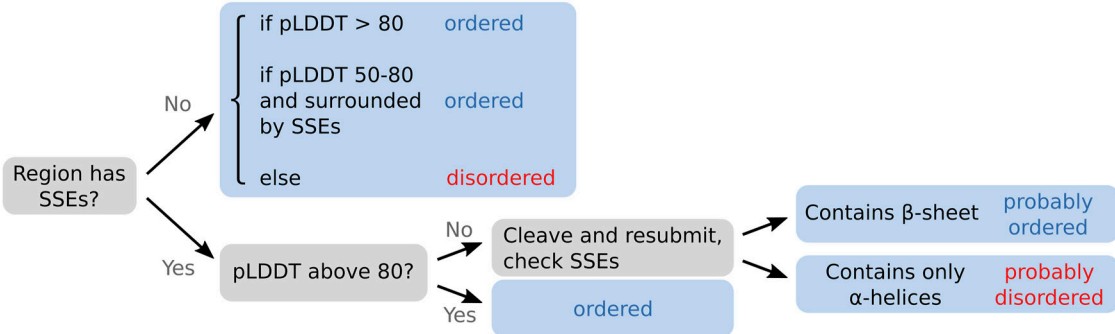

**Figure 2. Decision tree for the classification of regions from fold predictions into ordered or disordered.**
SSEs stand for "secondary structure elements"; pLDDT stands for "predicted local distance difference test." Multiple iterations are only required if SSEs with low pLDDT or large low pLDDT loops at the termini are present, where cleavage must take place.

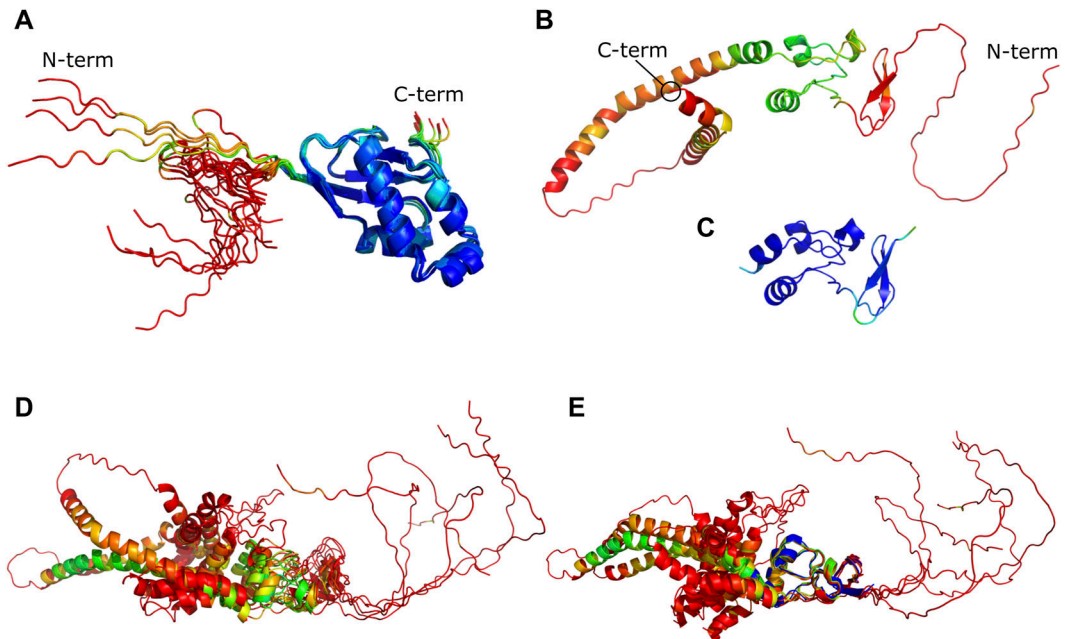

**Figure 3. Ordered and disordered region in relation to confidence scores in predicted models.**
Residues are colored according to their pLDDT, with blue representing high confidence and values above 90, red representing low confidence with values below 50, and other colors representing values in between. Structures within the upper and the lower panel share the same scales, respectively. **(A)** Ensemble of 20 predicted models emphasizing the difference between disorder (red, left) and order (blue, right). Predictions of the Ubl1 domain from SARS-CoV-1 via AlphaFold2 by running five models for a total of four seeds. Although the secondary structure elements overlap clearly, the disordered N terminus to the left varies drastically between models. **(B)** Prediction of complete βSM domain from SARS-CoV-2 with low-confidence secondary structure elements and large disordered termini. **(C)** Prediction of a cropped βSM domain sequence lacking the termini, showing an increased overall pLDDT. **(D)** Ensemble of five predicted models aligned to the rank 1 model, with the central fold recognizable. **(E)** Same ensemble as in (D) aligned to the experimentally determined structure 7T9W colored in blue. Disordered regions, despite being predicted as helices, point away in various directions.

During the classification into ordered and potentially disordered regions, few domains were divided into new subdomains and linkers, which extend the number of ranges compared with previous literature. One such case is the division of ßSM into ßSM-N (N-terminal linker), ßSM-M (ordered fold), and ßSM-C (C-terminal linker). Although the terminal linkers ßSM-N and ßSM-C are likely disordered because of low sequence identity in alignments, which is also reflected in low pLDDT values, ßSM-M resembles the experimentally determined structure 7T9W. As we show in Section B,

the domain between PL2^pro and NAB is betacoronavirus-specific, hence the name "betacoronavirus-specific linker domain."

The final domain ranges of all three viruses were compared for sequence similarity and in addition for similarity between predicted structures by calculating their RMSD values (Table 3). The predictions for both sarbecoviruses show high similarity with all RMSD values below 0.8 Å (excluding the transmembrane domains). Comparing both viruses with MHV, however, only seven out of 11 domains (excluding disordered and transmembrane domains)

**Table 2. Ranges of all domains and linkers determined by combining AlphaFold2 and experimentally solved structures.**

| Complete name | Abbreviation | Amino acid residue ranges | | |
|---|---|---|---|---|
| | | SARS-CoV-2 | SARS-CoV-1 | MHV |
| N-terminal loop of Ubl1 | Ubl1-N | 1–16 | 1–17 | 1–16 |
| Ubiquitin-like domain 1 | Ubl1 | 17–111 | 18–107 | 17–113 |
| Hypervariable region | HVR | 112–208 | 108–186 | 114–272 |
| Papain-like protease 1 | PL1pro | — | — | 273–476 |
| Linker PL1pro-Mac1 | | — | — | 477–487 |
| Macrodomain 1 | Mac1 | 209–377 | 187–355 | 488–644 |
| Linker Mac1-Mac2 (*Sarbecovirus*)/linker Mac1-DPUP-like (MHV) | | 378–412 | 356–390 | 645–666 |
| Linker Mac1-DPUP-like helix | | – | – | 667–680 |
| Linker pre DPUP | | — | — | 681–703 |
| Macrodomain 2 | Mac2 | 413–540 | 391–517 | — |
| Linker Mac2-Mac3 | | 541–550 | 518–526 | — |
| Macrodomain 3 | Mac3 | 551–675 | 527–651 | — |
| Domain preceding Ubl2 and PL2pro/DPUP-like domain (MHV) | DPUP | 676–745 | 652–722 | 704–777 |
| Ubiquitin-like domain 2 | Ubl2 | 746–804 | 723–781 | 778–837 |
| Papain-like protease 2 | PL2pro | 805–1,056 | 782–1,036 | 838–1,084 |
| betacoronavirus-specific linker domain | βSLD | 1,057–1,090 | 1,037–1,067 | 1,085–1,116 |
| Linker ßSLD-NAB | | — | — | 1,117–1,135 |
| Nucleic acid–binding domain | NAB | 1,091–1,196 | 1,068–1,174 | 1,136–1,211 |
| Betacoronavirus-specific marker domain N-terminal subdomain | ßSM-N | 1,197–1,239 | 1,175–1,217 | 1,212–1,292 |
| Betacoronavirus-specific marker domain Folded Core | ßSM-M | 1,240–1,325 | 1,218–1,304 | 1,293–1,369 |
| Betacoronavirus-specific marker domain C-terminal subdomain | ßSM-C | 1,326–1,412 | 1,305–1,390 | 1,370–1,448 |
| Transmembrane domain 1 | TM1 | 1,413–1,435 | 1,391–1,413 | 1,449–1,471 |
| Linker TM1-Ecto | | 1,436–1,442 | 1,414–1,419 | 1,472–1,504 |
| Ectodomain core fold | EctoC | 1,443–1,477 | 1,420–1,453 | 1,505–1,534 |
| Ectodomain linker/Linker Ecto-TM2 (MHV) | EctoL | 1,478–1,499 | 1,454–1,475 | 1,535–1,564 |
| Ectodomain TM-like helix | EctoTM | 1,500–1,522 | 1,476–1,492 | — |
| Linker EctoTM-TM2 | | 1,523–1,531 | 1,493–1,495 | — |
| Transmembrane domain 2 | TM2 | 1,532–1,554 | 1,496–1,518 | 1,565–1,587 |
| Linker TM2-AH1 | | 1,555–1,560 | 1,519–1,522 | 1,588–1,607 |
| Amphipathic helix 1 | AH1 | 1,561–1,583 | 1,523–1,545 | 1,608–1,630 |
| Linker AH1-Y1 | | 1,584–1,598 | 1,546–1,575 | 1,631–1,660 |
| Nidovirus-conserved domain of unknown function | Y1 | 1,599–1,759 | 1,576–1,736 | 1,661–1,819 |
| Y1 subdomain a | Y1a | 1,599–1,664 | 1,576–1,646 | 1,661–1,731 |
| Y1 subdomain b | Y1b | 1,665–1,759 | 1,647–1,736 | 1,732–1,820 |
| Linker Y1-CoV-Y | | 1,760–1,765 | 1,737–1,742 | 1,821–1,824 |
| Coronavirus-specific C-terminal domain | CoV-Y | 1,766–1,945 | 1,743–1,922 | 1,825–2,006 |
| CoV-Y subdomain a | CoV-Ya | 1,766–1,847 | 1,746–1,824 | 1,825–1,908 |
| CoV-Y subdomain b | CoV-Yb | 1,848–1,945 | 1,825–1,922 | 1,909–2,006 |

Shaded entries are predicted to have a defined structure, whereas nonshaded entries describe predicted regions of disorder. Domains discussed in more detail are shaded in red. Note that these domain boundaries are based on predicted order and are useful for crystallization trials. For full biological functionality, some domains may require their surrounding linkers.

**Table 3. Sequence similarities and RMSD values between pairs of MHV, SARS-CoV-1, and SARS-CoV-2.**

| Domain | SARS-CoV-2 to SARS-CoV-1 | | SARS-CoV-2 to MHV | | SARS-CoV-1 to MHV | |
|---|---|---|---|---|---|---|
| | Sequence similarity | RMSD | Sequence similarity | RMSD | Sequence similarity | RMSD |
| Y1 | 97.5% | 0.1 Å | 65.8% | 0.7 Å | 65.0% | 0.7 Å |
| Ubl2 | 97% | 0.1 Å | 54% | 0.6 Å | 52% | 0.6 Å |
| CoV-Y | 96.5% | 0.3 Å | 56.5% | 2.1 Å | 57.3% | 2.4 Å |
| βSLD | 90% | 0.3 Å | 54% | 0.2 Å | 54% | 0.4 Å |
| Ubl1 | 90% | 0.5 Å | 53% | 0.4 Å | 55% | 0.6 Å |
| Mac3 | 89.6% | 0.2 Å | — | — | — | — |
| PL2$^{pro}$ | 89.3% | 0.3 Å | 47.1% | 1.1 Å | 48.2% | 1.0 Å |
| NAB | 88.7% | 0.3 Å | 45.6% | 1.0 Å | 42.6% | 0.9 Å |
| ßSM-M | 88% | 0.7 Å | 42% | 3.4 Å | 39% | 2.9 Å |
| DPUP | 87% | 0.3 Å | 50% | 2.9 Å | 60% | 3.2 Å |
| ßSM-N | 86% | — | 53% | — | 60% | — |
| Mac1 | 85.5% | 0.2 Å | 52.6% | 0.7 Å | 51.7% | 0.8 Å |
| Mac2 | 84.9% | 0.4 Å | — | — | — | — |
| EctoC | 77% | 0.5 Å | 50% | 7.2 Å | 48% | 8.6 Å |
| ßSM-C | 75% | — | 63% | — | 46% | — |
| HVR | 62% | — | 35% | — | 35% | — |
| TM1 | 82% (22) | 0.3 Å | 67% (6) | 0.2 Å | 40% (5) | 0.2 Å |
| EctoL + EctoTM | 87% (38) | — | 62% (13) | — | 69% (13) | — |
| EctoTM | 75% (16) | 0.1 Å | — | — | — | — |
| TM2 | 80% (10) | 1.7 Å | 83% (6) | 0.7 Å | 75% (8) | 0.3 Å |
| AH1 | 100% (8) | 0.5 Å | 48% (21) | 0.5 Å | 53% (17) | 0.5 Å |

Only domains predicted to fold into a defined structure and large regions of disorder are listed. Short linkers are omitted. RMSD values are calculated for defined folds with PyMOL (31). Results are sorted by decreasing sequence similarity between both sarbecoviruses. Domains of the transmembrane region are listed below and are not sorted, because only short alignments were found. For these cases, the alignment length is given in parentheses.

show high similarity with RMSDs of 1.1 Å or less. With RMSD values above 7 Å, the prediction of MHV's ectodomain stands out, which is explored in Section C (Fig S4).

In contrast to high structural similarity between MHV and both sarbecoviruses, the sequence similarities are relatively low with values from 42.6% to 65.8%. The C-terminal domains Y1 and CoV-Y stand out with high sequence similarities of up to 97.5% despite containing more than 150 residues, indicating an important function. Surprisingly, the linker domain is listed at position 4 with higher similarity than PL2$^{pro}$ and the macrodomains. Its structure and conservation among coronaviruses are explored in the next section.

### Experimental validation of the betacoronavirus-specific linker domain (βSLD)

#### Structure prediction of linker domain between PL2$^{pro}$ and NAB domain

The linker between PL2$^{pro}$ and NAB stands out because of its high sequence similarity of 90% between SARS-CoV-2 and SARS-CoV-1, higher than the similarities of most other folded domains (Table 3). In addition, AlphaFold2 predicts a stable fold consisting of

34 residues for SARS-CoV-2, forming two β-hairpins connected by an eight-residue loop (Fig 4). This loop was predicted to form a helix-like structure, which could potentially form because of a restriction in its conformational space induced by Pro1076 and two hydrogen bonds within its backbone. A similar helix-like structure is found at the N terminus, also containing a proline and two internal hydrogen bonds. The fourth β-strand in the sequence Nterm-β1-β2-loop-β3-β4-Cterm is connected via two hydrogen bonds to β1 in parallel direction, leading to a compact fold. Strand β1 is connected to β2 via hydrogen bonds in the backbone and in addition via the Thr1063-Thr1072 and Tyr1064-Glu1073 hydrogen bonds (Fig 4). Ramachandran analysis via MolProbity (33) shows only Thr1058 as an outlier, whereas 26 out of 32 residues were in favored regions (Ramachandran plots in Fig S2).

#### Conservation of the linker domain

The sequence of this linker domain from MHV has a high sequence similarity to the linker domain from both sarbecoviruses (54%). A BLAST search yielded additional hits within *Betacoronavirus*, but no hits outside of this genus.

Local pairwise and multiple sequence alignments between the sequence from SARS-CoV-2 and sequences from 16 betacoronaviruses

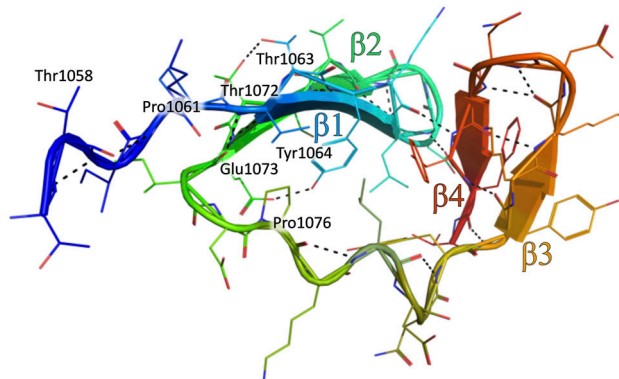

**Figure 4. AlphaFold2 structure prediction of the SARS-CoV-2 nsp3 linker domain.**
N terminus is colored in blue, and C terminus in red. Hydrogen bonds are depicted as black dotted lines. The labeled prolines potentially restrict the conformational freedom of their loops, with Pro1076 being located in the central loop. The other labeled residues interact via hydrogen bonds with each other. Residue numbers are based on SARS-CoV-2 nsp3 sequence.

revealed conserved residues (Fig 5), namely, Leu1066, Asp1067, Pro1076, Tyr1088, and Thr1090 (residue numbers based on SARS-CoV-2 nsp3). Furthermore, the residues 1,064, 1,081, and 1,089 are always Phe or Tyr; Leu1078 is also mostly conserved. From 34 residues, 14 retain similar chemical properties, including fully conserved ones (Fig 5). From those, 12 are in a loop or at the edge of a β-strand right next to a loop, indicating a high selection pressure on the loop regions. The structure predictions show in all cases a highly similar fold with RMSD values below 0.2 Å.

Additional sequence alignments with the unclassified shrew coronavirus, alphacoronaviruses (24 viruses), gammacoronaviruses (5 viruses), and deltacoronaviruses (10 viruses), revealed no hits within the region of nsp3. However, several hits were observed in the region of endoRNase, known as nsp15 in SARS-CoV-2. Structure predictions reveal a similar fold as for the linker domain, but with an additional helix before the third β-strand. Details are in Supplemental Data 1 Section B Table S3. Because no sequence homologs were identified outside of *Betacoronavirus*, we named this domain betacoronavirus-specific linker domain.

### Experimental validation

To assess whether the betacoronavirus-specific linker domain (βSLD) is in fact folded, single-crystal X-ray diffraction and small-angle X-ray scattering (SAXS) experiments were conducted. Because of the small size of the linker domain and the high-confidence prediction of a close arrangement between this domain and PL2$^{pro}$, a multidomain construct was designed. The construct comprised SARS-CoV-2 residues Thr1057 to Thr1090, including the domains Ubl2, PL2$^{pro}$, and βSLD. Furthermore, the PL2$^{pro}$ C111S mutant was used for better crystallization chances (16).

Crystallization trials led to thin crystals, from which data could be collected and processed into an electron density map. Unfortunately, the model building revealed that cleavage took place and only half of the construct assembled into the

crystal. It was the undesired half covering Ubl2 and approximately half of PL2$^{pro}$. Despite the surprising result that half of PL2$^{pro}$ crystallizes, no useful information about the structure of βSLD could be gathered. It is unclear whether PL2$^{pro}$ was involved in the proteolysis or other factors introduced a systematic cleavage. However, it is unlikely that PL2$^{pro}$ is able to self-cleave in vivo because of its sequence specificity (4) and the assembly of nsp3 into large hexameric complexes (7), as well as no reports about such behavior. Analysis of the construct before crystallization showed a band at the expected size. Also, remaining protein sample was used in a SAXS experiment, where cleavage of the construct could be excluded.

The SAXS result shows a good agreement between the relaxed AlphaFold2 prediction and the experimental data, with a $\chi^2$ of 0.98 (Fig 6A) (the relaxed AlphaFold2 prediction is the most fitted structure found by the end of a single SREFLEX (34) run). The dimensionless Kratky plot suggests that the solution structure of Ubl2-PL2$^{pro}$-βSLD is a rather compacted multidomain entity (Fig 6B). It is highly unlikely that a long disordered tail exists, according to the flat plateau at the high scattering vector in the Kratky plot; instead, short flexible regions between domains are expected. The relaxed AlphaFold2 prediction also fits well against the envelope of the ab initio SAXS model (Fig 6C).

### Nucleic acid–binding domain and betacoronavirus-specific marker domain

The subsequent domains from the betacoronavirus-specific linker domain, NAB (nucleic acid–binding domain) and βSM (betacoronavirus-specific marker domain), are also betacoronavirus-specific (4, 17). However, a region similar to βSM was identified in gammacoronaviruses, known as γSM (gammacoronavirus-specific marker domain) (4). Because of the large, disordered termini of βSM, we introduced a more specific nomenclature, with βSM-N (N-terminal link), βSM-M (folded central unit), and βSM-C (C-terminal link) with ranges in Table 2.

Sequence alignments within *Betacoronavirus* and structure predictions show that both, NAB and βSM-M, are present in all betacoronaviruses. Alignments between sequences from SARS-CoV-2 and 15 betacoronaviruses, excluding SARS-CoV-1, show sequence identities from 27% to 38% for NAB and 18% to 33% for βSM-M. NAB and βSM-M show also a high sequence similarity of 56% in an alignment of 25 residues, which could serve as a hint on their evolution.

In *Gammacoronavirus*, alignments with the Canada goose coronavirus showed the highest sequence identities, with 22% for NAB and 31% for βSM-M. Structure prediction from an extended βSM-M sequence based on that alignment led to a high-confidence structure prediction closely resembling SARS-CoV-2 βSM-M (RMSD 1.5 Å). However, no other gammacoronavirus showed similar results. Nevertheless, Canada goose coronavirus could serve in further experiments as proof that βSM-M is not completely betacoronavirus-specific as are NAB and the betacoronavirus-specific linker domain.

### Insights into the predicted hexameric assembly of Y1+CoV-Y

The C-terminal region of nsp3, containing the transmembrane domains and the lumenal ectodomain, as well as the Y1 and CoV-Y

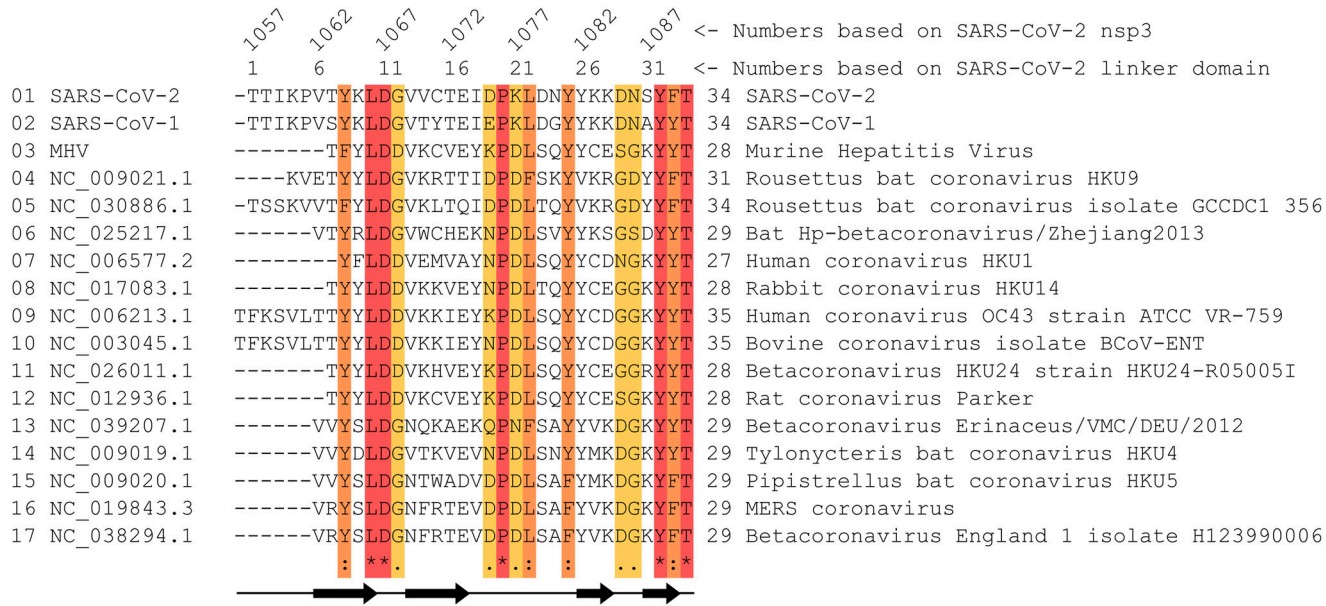

**Figure 5. Multiple sequence alignment between the linker domain of various betacoronaviruses.**
The sequences for the linker domains were first identified with a local pairwise sequence alignment between the linker domain sequence of SARS-CoV-2 and orf1ab sequence of the respective virus. Afterward, the identified sequences were used in a global multiple sequence alignment, resulting in this figure. Residues conserved in all examined viruses are shown in red, residues with strongly similar chemical properties at the same position are highlighted in orange, and weakly similar properties are shown in yellow. The secondary structure elements are indicated at the bottom.

domains, is only partially covered by two PDB entries: 7RQG showing the C-terminal subdomain of CoV-Y (CoV-Yb) (18); and a larger construct (8F2E) covering Y1b and CoV-Y (19) (unfortunately, the later study (19) refers to a portion of CoV-Y as "Y3," which differs from the subdomain labeled "Y3" in the PDB entry 7RQG. To resolve this conflict, we designated these subdomains in Table 2 as "Y1a," "Y1b," "CoV-Ya," and "CoV-Yb").

Y1 is referred to as "nidovirus-conserved domain of unknown function," whereas CoV-Y is a C-terminal nsp3 domain specific to *Coronaviridae* (17). The lumenal ectodomain was shown to interact with nsp4, resulting in the formation of the viral replication organelles known as "double-membrane vesicles" (6). The cytosolic domains of nsp3 assemble into a hexameric complex participating in the export of new viral RNA genomes (7).

We analyzed the structure prediction of Y1 and CoV-Y and investigated the conservation of Y1 among related viruses. Furthermore, we analyzed the prediction of a Y1 hexamer.

### Structure prediction of Y1 and CoV-Y

For both sarbecoviruses and MHV, a folded structure was predicted by AlphaFold2 for the region covering Y1 and CoV-Y (Fig 7A), consisting of 362–377 residues. The predicted model consists of an N-terminal helical loop with low pLDDT, a high-confidence fold of ~160 residues, a short low pLDDT loop, and another high-confidence fold (Fig 7A). The first ~180 residues correspond to the domain Y1, whereas the remaining ones are part of CoV-Y.

The predicted alignment error (pAE) matrix suggests a clear separation between Y1 and CoV-Y at Gly1763 (SARS-CoV-2 nsp3), but no further division into subdomains (Fig 7B). However, both domains consist of two distinct globular folds, allowing a separation

of the subdomains Y1a and Y1b (Fig 7D), as well as CoV-Ya and CoV-Yb (Fig 7A), which is supported by the experimental structure 8F2E. Only Y1a has not yet been experimentally determined. Repeated predictions of this domain pair show similar results with the same orientation of both domains relative to each other in all cases, but the pAE matrix indicates a potential flexibility in the arrangement and orientation of CoV-Y to Y1.

The fold of Y1a comprises a large β-hairpin (21 residues in length), a 50-residue region of loops, and an 11-residue-long α-helix (Fig 7D). This globular fold consisting mostly of loops is held together by numerous hydrogen bonds and contains a conserved cysteine–histidine cluster, which could stabilize the fold by binding metal ions (Fig 7C).

Y1b makes up the upper half of Y1 (90 residues), containing a parallel β-sheet made of three β-strands, which creates an intertwined structure with a triplet of antiparallel β-strands, an α-helix, a loop stabilized by H-bonds, and sharp turns.

The subdomain CoV-Ya consists of four α-helices and CoV-Yb of four α-helices, a β-sheet made of four β-strands, and various loops and turns, resulting in a globular fold (Fig 7A). CoV-Yb resembles closely the PDB structure 7RQG (18) with an RMSD of 0.4 Å. Because Y1a is missing in the PDB entry 8F2E (19), the domains in the prediction are differently packed, resulting in a high RMSD of 2.3 Å. Aligning only the individual subdomains from 8F2E to the prediction in Fig 7A, however, results in RMSD values of 0.4 Å for all three subdomains.

A second potential zinc site, consisting of a histidine and three cysteines, was identified in the linker between AH1 (amphipathic helix 1; the domain preceding Y1) (35). Because the pAE suggests no alignment between N terminus and the main fold of Y1, it was

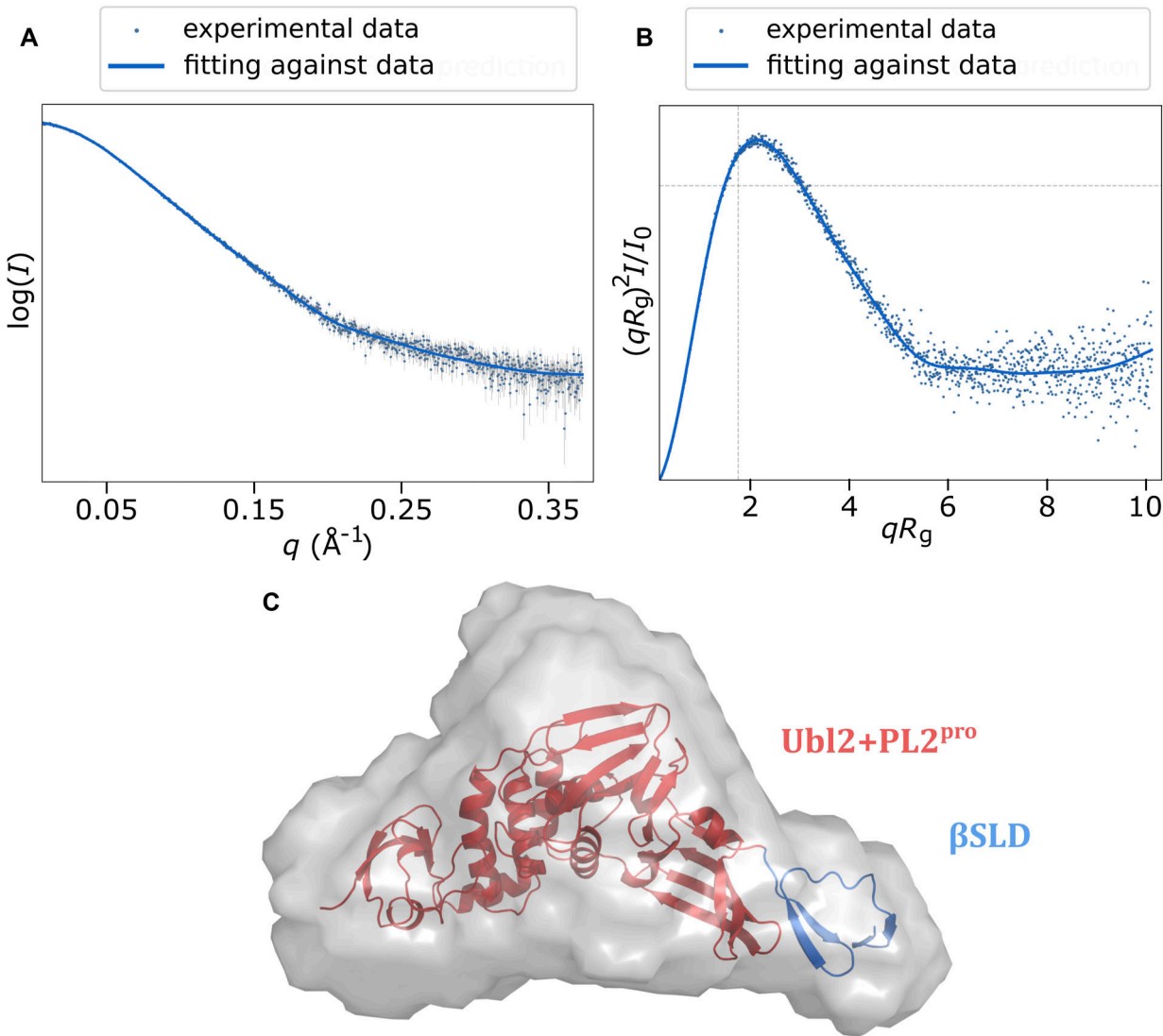

**Figure 6. Experimental solution SAXS result of Ubl2-PL2$^{pro}$-$\beta$SLD.**
**(A)** Fitting result of the relaxed AlphaFold2 model against the experimental SAXS data. Dots are the SAXS data with the relative errors. Solid line is the estimated scattering curve of the relaxed AlphaFold2 prediction. **(B)** Dimensionless Kratky plot. The peak maximum largely shifts away from the theoretical value for a compacted globular fold (marked by the two dashed gray lines) suggests the solution structure of Ubl2-PL2$^{pro}$-$\beta$SLD is a multidomain protein. However, the convergence of the Kratky plot at higher $q$ means there are no long flexible linkers between domains or at the N, C termini of the entity. **(C)** Projection of the envelope of the ab initio model (gray volume) and the relaxed AlphaFold2 prediction (cartoon), with the linker domain at the right side in blue and the remaining part being Ubl2+PL2$^{pro}$ in red.

excluded from the domain boundaries of Y1 and was left as a linker (Table 2). Because of the potential metal-binding cluster, however, it could belong to the amphipathic helix or Y1, which would depend on its function. In SARS-CoV-2, this situation is more complicated, as the linker is shorter and the histidine of the zinc-binding site is located in AH1. Without additional information regarding this zinc-binding site's function, it is not possible to define better domain boundaries.

### Conservation of Y1 among nidoviruses
Despite long sequences of more than 360 residues, the C-terminal domains Y1 and CoV-Y exhibit high sequence similarities between the two sarbecoviruses and MHV, ranging from 65% to 97.5% for

Y1 and from 56.5% to 96.5% for CoV-Y (Table 3). Structure predictions show nearly identical folds for Y1 with RMSDs from 0.1 to 0.7 Å; CoV-Y shows less similarity with RMSDs from 0.3 to 2.4 Å.

Because Y1 is also known as "nidovirus-conserved domain of unknown function" (17), the sequence and structure similarities to other viruses were investigated: first, with other betacoronaviruses, followed by other viruses from *Orthocoronavirinae*, close relatives from *Coronaviridae*, and related nidoviruses.

The global sequence alignment of SARS-CoV-2 with 16 betacoronaviruses (listed in Fig 5) revealed 29 out of 161 amino acids to be conserved across all 17 viruses (Fig 7D; details in Supplemental Data 1), with 22 residues being in Y1a. Further 45 residues of Y1 show conserved chemical properties. The 22 conserved residues in Y1a

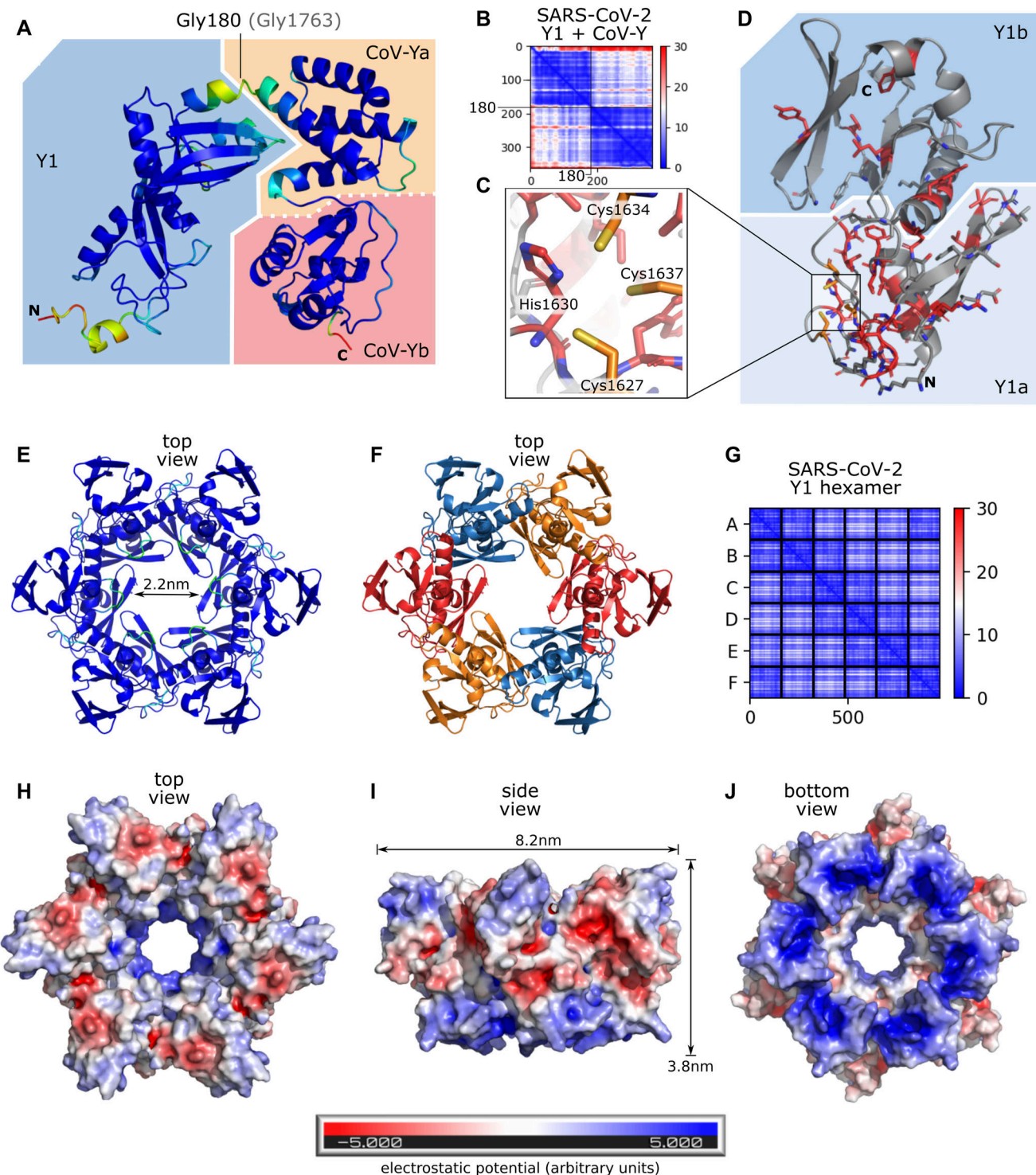

**Figure 7. Prediction of the nsp3 domain Y1 as monomer and hexamer.**
**(A, B)** Prediction of linker + Y1+CoV-Y from SARS-CoV-2 (ranges from Table 2) colored according to pLDDT with blue as high confident (A) with the respective pAE matrix (B). **(C, D)** Matrix shows two confident local folds, one with residues before Gly180 and one with residues after it; (C) prediction of conserved cysteine–histidine cluster; (D) Y1 according to the ranges in Table 2, monomer from hexamer prediction, with amino acids which are conserved among all betacoronaviruses highlighted in red, and the cysteine cluster from (C) in orange. **(E, F, G)** Other shown residues are involved in hexamer formation but are not conserved; (E, F, G) high-confidence prediction of Y1 hexamer colored according to pLDDT (E) and by chain (F). The narrowest diameter of the inner gap is 2.2 nm. **(H, I, J)** The pAE matrix shows a highly confident arrangement of all monomers (G); (H, I, J) Y1 hexamer with electrostatic surface calculation via PyMOL (31) viewed from different angles. Bottom surface (J) and the surface along the channel (H, J) are positively charged (blue), whereas the top surface (H) and side surface (I) are mostly neutral (white) or negatively charged (red).

include a potential zinc-binding site (35) made of three cysteines and a histidine (Fig 7C). With distances of 3.4 Å to 4 Å, the cysteines are too far apart for disulfide bond formation. Furthermore, the conserved amino acids often form hydrogen bonds with each other, which potentially stabilizes the loop region of Y1a.

Across all examined coronaviruses outside *Betacoronavirus*, Y1 shows moderate sequence similarity from 25.9% to 39.1%, but exhibits high structural similarity with RMSD from 0.6 Å to 1.3 Å in comparison with structure predictions. The closest relatives outside *Orthocoronavirinae* are found within *Pitovirinae*, *Letovirinae*, *Roniviridae*, *Mesoniviridae*, *Arteriviridae*, and *Tobaniviridae* (17). However, despite showing partially confident fold predictions at sequence identities above 20%, none of the predicted structures resembled the coronaviral Y1, thus questioning Y1's conservation among nidoviruses (details on used species are in Supplemental Data 1).

### Multimer prediction of Y1 and Y1+CoV-Y

The multimer feature of AlphaFold2 (36 *Preprint*) and ColabFold (37 *Preprint*) predicts the assembly of homo- and heteromers. It generates a special pAE matrix that evaluates the distances between residues within a monomer and residues of different monomers, thereby assessing the confidence of the predicted complex (Fig 7G).

Nsp3 is forming a hexameric complex with nsp4 and nsp6 (7). From all domains, only the prediction of hexameric Y1 (Fig 7E and F) comes with a confident arrangement of all monomers (Fig 7G). The average pLDDT of the multimer remains highly confident with a reduction of only 1.4 compared with the monomer. The average pAE per cell of the pAE matrix (Fig 7G) ranges from 3.2 to 8.4, which is in the confident range of below 10. The low values in the pAE matrix and high pLDDT are consistent across multiple predictions and for all examined viruses. Furthermore, the prediction of the Y1 hexamer and the Y1+CoV-Y hexamer shows no Ramachandran outliers.

The hexameric assembly forms a channel with a minimum inner diameter of 2.2 nm. The monomers are held together by several hydrogen bonds, which are formed in SARS-CoV-2 nsp3 by the 11 residues Arg1602, Thr1607, Gly1611, Ser1615, Ala1642, Asp1654, Leu1718, Ser1729, Ser1734, Lys1737, and Leu1746.

Vacuum electrostatics were calculated for the Y1 hexamer (Fig 7H–J) in PyMOL (31). These show a primarily positive charge at the inner surface of the channel and on the bottom surface toward the membrane. The sides and the top surface show slightly positive or negative charge areas, as well as some neutral areas and few highly positive charges in the buried region.

The multimer prediction of SARS-CoV-2 Y1+CoV-Y leads to a similar complex as solely with Y1. Although the Y1 domains remain assembled with high confidence in the pAE matrix, CoV-Y domains show low confidence in the pAE for the alignment between each other and the arrangement to their linked Y1. For MHV, however, the pAE matrix shows much better arrangement, although not such a perfect one as for Y1 in Fig 7G (see Fig S3).

In UCSF Chimera (38), the Y1-CoV-Y hexamer with lowest pAE values, which was predicted from MHV, was fitted into the cryoelectron tomography map from reference 7. This 30.5 Å resolution map shows the whole hexameric pore complex consisting of nsp3,

nsp4, and nsp6, and covers also the membrane of the replication organelle. The fitting algorithm finds a position for the model at the base of the pore complex (Fig 8). The channel diameter of the model matches that of the map (Fig 8A–C). From the base, six pillars grow up, where the CoV-Y domains are fitted in (Fig 8D–G). Because of the low resolution, parts of the model are not covered in the volume. Different contour levels change this, but obscure also the channel.

The hexameric prediction of Y1+CoV-Y is not as confident in pLDDT and pAE as the prediction of hexameric Y1, but in the case of MHV, the prediction confidence improves in contrast to the prediction of the Y1+CoV-Y monomer. Despite the uncertainty in the arrangement of both domains (in the pAE matrix), the predicted model fits well into the cryoelectron tomography map.

To form replication organelles, the ectodomain of nsp3 must interact with nsp4, which is also part of the hexameric complex. Our results revealed a globular fold within the ectodomain and a large ectodomain in nsp4 (Fig S4, Table S4). However, all multimer predictions of the latter domain contradict the experimental evidence and come with low-confidence scores.

The expression of a construct containing the Y1 domain was attempted in BL21Gold *E. coli*. However, insufficient amounts of the desired protein were synthesized, preventing an experimental validation of the hexameric Y1 complex. Other, more sophisticated expression hosts (e.g., insect or mammalian cells) might be required to produce useful quantities of Y1.

## Discussion

Sequence-based structure prediction is transforming the field of structural biology. Although AlphaFold2 predictions are not as reliable as experimental measurements, they can give a good indication of intrinsic disorder and are well suited to assist construct design (29). We used AlphaFold2 to determine domain boundaries more exactly, identified a new domain directly adjacent to the most important nsp3 drug target PL2$^{pro}$, and established a potential mechanism driving the hexameric assembly of the pore.

### Using AlphaFold2 for domain boundary determination and construct design

#### Classification into regions of order and disorder

Correctly predicting secondary structure elements is crucial to differentiate between regions of order and disorder and for the design of crystallizable constructs. Superimposing the AlphaFold2 predictions with experimentally determined domain structures showed that all secondary structure elements were correctly predicted. For all but two, root mean square deviation (RMSD) values were below 1 Å. The two exceptions were Ubl1 from SARS-CoV-1 and MHV with the NMR structures 2GRI and 2M0A, which have a flexible N terminus. Despite high RMSD values in these cases, the folded region was predicted correctly.

AlphaFold2 predicts disordered loops with low pLDDT values and stretches them away from folded regions, forming the barbed-

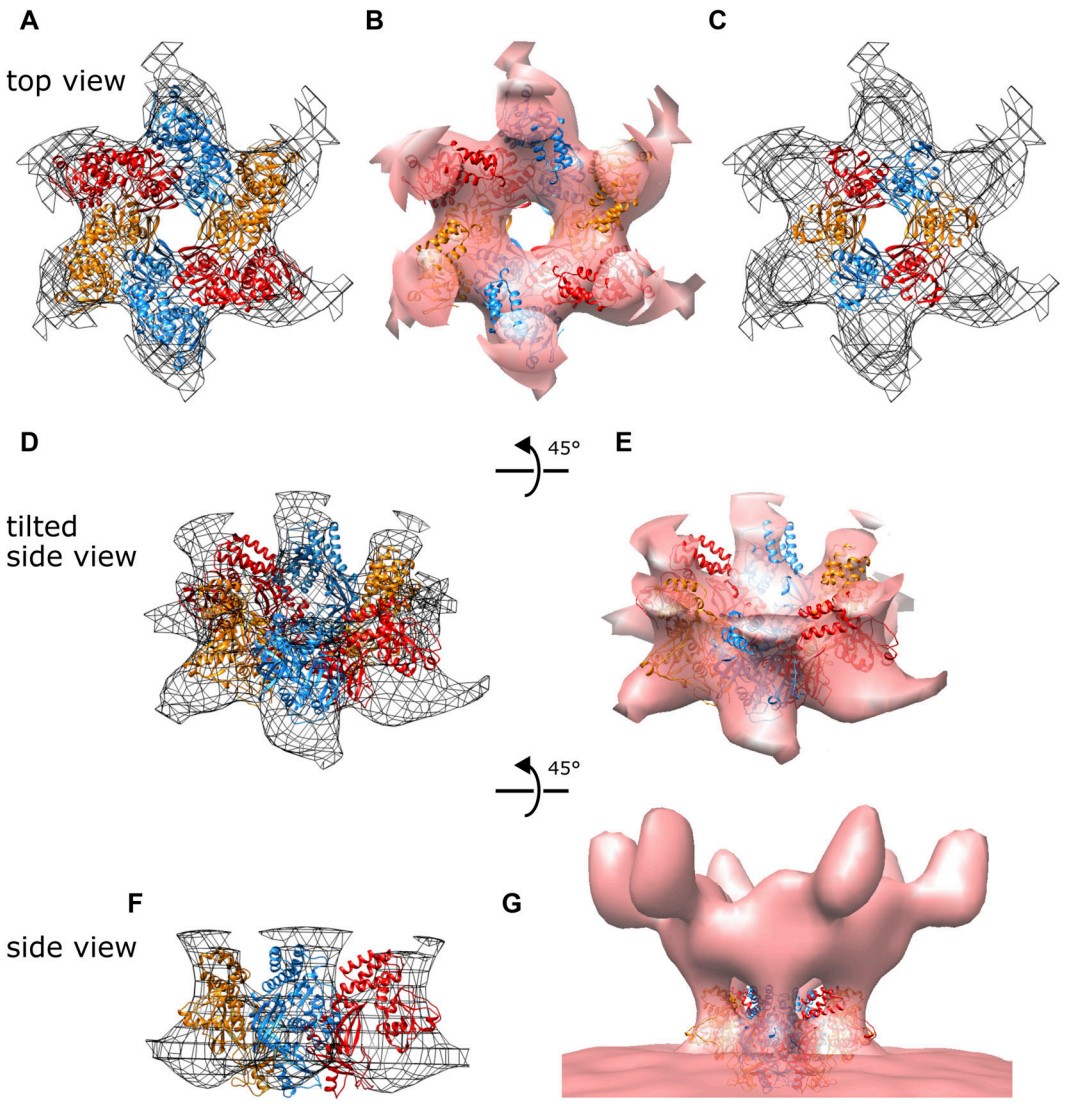

**Figure 8.  Fit of the predicted Y1 hexamer structure into cryoelectron tomography map.**
**(A, B)** Top view of the predicted model of Y1+CoV-Y hexamer from MHV fitted into cryoelectron tomography map from reference 7, with volume shown as grid (A) and surface (B). **(C)** Same fit, but only Y1 hexamer is shown. **(D, E)** Tilted side view of same model fit as in (A, B) with camera angle rotated by 45°. **(F)** Side view of same model fit as in (D) with camera angle rotated by 45°. For more clarity, only half of the hexamer is shown. **(G)** Side view of the model fitted into the map with the whole pore complex visible. All images were generated in UCSF Chimera (38) with a contour level of 2.89. For images (A, B, C, D, E, F), volume was truncated to highlight the complex's base.

wire–like conformation first described by reference 28. If these disordered regions are large, they can negatively impact the pLDDT of nearby ordered regions and lead to folds differing from the experimentally determined structure, as in the example of the βSM domain. Furthermore, disordered regions with low pLDDT values were sometimes falsely predicted as α-helices. In our cases, such bias could be detected by truncating the low pLDDT loops from the input sequence. This increased the pLDDT of the ordered fold drastically, whereas the pLDDT of false helices decreased. We also aligned multiple models predicted from the same sequence and observed here good 3D alignment at the ordered sections, whereas disordered loops and false helices pointed away from the folds in random directions, as illustrated by Fig 3D and E. Transmembrane

helices were in our cases also predicted to point away from the main fold in random directions, but they differed from the false helices by having pLDDT values above 90. Because false β-sheets were not observed, the presence of predicted β-sheets could be a good indicator of order.

It is important to note that pLDDT calculation depends on the coverage of the sequence during the multiple sequence alignment step of AlphaFold2 (22). Sequences with a low number of similar sequences in the database are thus predicted with low pLDDT values. The minimum numbers of similar sequences for ensuring our method to work reliably are yet unclear. In our minimum case, we achieved high pLDDT values (above 80 for most of the residues) with a coverage by 14 sequences with sequence identities from

50% to 100% (for DPUP-like domain from MHV). In a case where sequence identities ranged from 20% to 100% (for SARS-CoV-2 Mac2), sequence coverage of 30 was sufficient.

Conclusively, AlphaFold2 predictions can be used to differentiate between order and disorder using the pLDDT, presence of predicted secondary structure elements, and the "barbed-wire" phenomenon (28). It is useful for determining domain ranges of multidomain proteins and designing crystallizable protein constructs by detecting disordered termini. Shortening the input sequences for AlphaFold2 across multiple runs can improve the confidence measurements and result in structures, which are closer to the experimentally determined structures. Nevertheless, these models remain predictions and are hence only a hypothesis-generating step in construct design, followed by experimental methods and construction of models based on physical data.

### Domain boundary determination

In a multidomain protein, experimental structure determination often requires understanding domain boundaries, in order to establish function and internal movement and to solve domains independently. Our method of domain boundary determination is based on AlphaFold2's capability to predict whether a region is ordered or disordered. During the preparation of short input sequences, covering only few or one domain, we used experimentally validated domain boundaries and aligned sequences from related viruses against sequences of established domains. Sequence alignments between homologs already can hint at order or disorder within a multidomain protein, as disordered regions show usually low sequence similarity compared with the similarity of folded domains (39). This is due to lower selection pressure in disordered regions, which hence accumulate mutations faster.

In our case, we aligned the individual domain sequences between both sarbecoviruses and mouse hepatitis virus. The folded domains have relatively high sequence similarities, whereas disordered regions such as linkers had lower similarity. The linker between PL2$^{pro}$ and NAB stood out with a high sequence similarity and was later found to be ordered. We suspect that this betacoronavirus-specific linker domain (further discussed in Section B) went unnoticed because of the incremental annotation of domains for nsp3 in the past. Probably, PL2$^{pro}$ and NAB domains were annotated first and the linker, as all other short linkers in nsp3, was not annotated as such explicitly. Hence, one intuitively assumes this region to be disordered, whereas in truth it is just not further explored. This case emphasizes the importance of clearly defined ranges and of annotations for disordered regions indicating whether they were experimentally proven not to be ordered, such as "unexplored," "predicted," or "experimentally validated."

Our goal was to map each residue in nsp3 to exactly one segment, which can be an individual ordered domain or a region of disorder between domains. Preliminary domain ranges were used as initial input to AlphaFold2. With the increased limit of ColabFold (37 Preprint) for sequences of up to 4,000 residues, predicting the complete structure of a multidomain protein is now also a good starting point for defining domain boundaries without any prior knowledge. In both cases, large regions of disorder can have a negative impact on the prediction of nearby folds as discussed in the previous section. Therefore, the input sequences must be refined by cropping disordered termini. After few iterations, we obtained the final domain ranges for nsp3 (Table 2). This is to our knowledge the first complete list with ranges for each domain and every linker in between of nsp3. However, experiments should be done to validate and complete this list. Until then, it serves as a starting point for construct design and discussion.

### Experimental validation of the betacoronavirus-specific linker domain (βSLD)

#### Structure analysis

Sequence alignments between SARS-CoV-1 and SARS-CoV-2 showed an unusually high sequence identity for the 34-residue linker located between PL2$^{pro}$, the most important drug target of nsp3, and NAB. AlphaFold2 predicted the domain as folded (Fig 4), and further sequence analysis showed this domain to be betacoronavirus-specific; hence, we called it betacoronavirus-specific linker domain (βSLD). Among 17 betacoronaviruses we aligned, five out of the 34 residues are conserved (Fig 5) and nine have similar properties (40), suggesting a functional role of this domain. From these 14 residues, 12 are located in a loop or at the edge of a β-strand right next to a loop (Fig 5). This high selection pressure suggests an important role of the loop residues in function or conservation of the fold.

The betacoronavirus-specific linker domain consists of two β-hairpins of two strands each (β1+ β2 and β3+ β4) (Fig 4). Both pairs are interconnected via two hydrogen bonds to each other, forming a compact and plausible fold. The 8-residue central loop is a potential weak point for structural stability but is predicted to form two hydrogen bonds within itself in its backbone. Its flexibility is further limited by a conserved proline, forming a helix-like structure element in the predicted model.

In conclusion, the predicted fold seems valid and structurally stable, although it has one disallowed Ramachandran outlier. Conserved residues overlap with the loop regions where specific residues could be important for maintaining the fold. Lastly, small-angle X-ray scattering (SAXS) measurements agree well with a relaxed model of the predicted structure. Therefore, this domain is very likely forming a stable fold.

#### Potential functions of the linker domain, NAB, and βSM

Although PL2$^{pro}$ is conserved across all coronaviruses and beyond, the three following domains, βSLD, NAB, and the betacoronavirus-specific marker domain (βSM), are specific to Betacoronavirus (4, 17). The βSM domain comprises an 80-residue-long disordered linker (βSM-N) between NAB and the folded part of βSM (βSM-M), as well as another disordered linker βSM-C connecting βSM-M with the transmembrane region.

Because NAB is able to bind DNA and RNA (4), it could support the export of new copies of the viral RNA genome, which would require a position near the channel of the hexameric RNA exporter pore (7). NAB is connected to the first of two transmembrane domains via βSM and both of its linkers (βSM-N and βSM-C). As Y1 follows the second transmembrane domain and does not have a large linker, its folded part must sit directly adjacent to the pore. Furthermore, the electron tomographic structure expands far into the cytosol (7), requiring most domains to sit on top of each other.

Because Mac2, Mac3, DPUP, Ubl2, PL2[pro], βSLD, and NAB are connected tightly (784 residues in SARS-CoV-2), with only Ubl1 and Mac1 as remaining cytosolic folded domains linked with higher flexibility, all those domains cannot be located adjacent to the pore. This leaves Y1 as a basis on the membrane with all remaining domains on top of it as the only plausible option. The βSM domain allows this due to its large disordered regions. In the next section, we go into more detail of Y1's role in this assembly.

To allow interaction between NAB and the viral RNA passing the channel, NAB must be located on top of Y1. However, NAB alone is too small to form a hexameric ring with the same diameter as reported from the cryoelectron tomography map (7), so it must assemble into a ring together with additional domains. βSLD and PL2[pro], as well as potentially βSM-M, could fill these gaps and position NAB toward the pore in a functional orientation. In such a scenario, βSLD and βSM-M would serve only a structural purpose. Mutation experiments with disrupted folds of these betacoronavirus-specific domains could give clues about their function in complex assembly and their impact on viral fitness. It is also possible that βSM-M and βSLD are regulating the activity of NAB, PL2[pro], and/or other enzymatically active domains, which appear close during complex assembly.

### Evolution of betacoronaviruses-specific domains

Because βSLD, NAB, and βSM are all specific to *Betacoronavirus* (4, 17) and potentially interact with each other during complex assembly, their coevolution could hint at their functions. To understand this coevolution, we analyzed the gammacoronavirus-specific marker domain (γSM). Only the prediction of Canada goose coronavirus γSM resembles the structure of βSM-M, whereas for all other gammacoronaviruses, other folds are predicted. Therefore, the fold of βSM-M is potentially common between both clades in coronaviruses, in contrast to the truly betacoronavirus-specific domains NAB and βSLD. In such a case, the virus would benefit from βSM-M without the presence of NAB or βSLD. However, this statement requires structural validation of the predicted fold of Canada goose coronavirus γSM and does not give new clues about the function of βSLD.

Also, we observed high sequence similarities between βSLD from SARS-CoV-2 and another nonstructural protein, endoRNase, in various alpha-, gamma-, and deltacoronaviruses. Structure predictions from these sequences have a similar fold to βSLD but with an additional helix between the central loop and the third β-sheet. A gene-duplication event might have occurred during betacoronavirus evolution, where part of endoRNase was duplicated and translocated to nsp3. Deletion of the helix residues allows in our structure prediction a compact and more stable fold. Nevertheless, this gives no information on whether βSLD evolved before or after, or coevolved with NAB, which would be interesting regarding βSLD's function.

### Insights into the predicted hexameric assembly of Y1+CoV-Y

The cytosolic C-terminal region of nsp3 right after the transmembrane region comprises the domains Y1 and CoV-Y (4, 7) (Fig 1B). Only recently, an experimental structure covering the second subdomain of Y1 and the complete CoV-Y domain was published (PDB entry 8F2E) (19). Sequence analysis combined with structure prediction enabled us to explore the potential function of the "nidovirus-conserved domain of unknown function" during nsp3 complex assembly.

### Nomenclature

Structure prediction of the complete C-terminal region (residues 1,584–1,945 in SARS-CoV-2 nsp3) shows a high-confidence fold comprising two large domains. Predicted alignment error (pAE) and low pLDDT values around Gly1763 suggest separation into two domains, named Y1 (residues 1,599–1,759) and CoV-Y (residues 1,766–1,945), with domain names according to reference 4. Both consists of two closely arranged globular folds and could therefore be divided into subdomains, which must be labeled according to structure and function. Unfortunately, older literature did not define the border between Y1 and CoV-Y, which led to conflict in nomenclature: the PDB structure 7RQG covers the last of four subdomains of Y1+CoV-Y (residues 1,844–1,945) and is published under the name Y3 (paper not published yet). The second experimentally solved structure from a recent publication (PDB 8F2E) (19) labels this subdomain Y4 and another domain (residues 1,764–1,847) Y3, making the name Y3 ambiguous.

In the publication of 8F2E (19), the C-terminal region is divided into four domains: Y1 (residue range not given), Y2 (residues 1,665–1,763), Y3 (residues 1,764–1,847), and Y4 (residues 1,848–1,945). These domain ranges closely resemble those from our results, namely, Y1a (residues 1,599–1,664), Y1b (residues 1,665–1,759), CoV-Ya (residues 1,766–1,847), and CoV-Yb (residues 1,848–1,945), respectively. However, the latter study suggests the three latter subdomains to be part of CoV-Y, whereas we suggest the first two subdomains to be part of Y1 and only the last two as part of CoV-Y, as explained below.

### Structure prediction of Y1+CoV-Y and domain separation

Domain ranges should generally be based on functional and tightly packed units. In the case of the C-terminal region of nsp3, we therefore consider Y1 with its subdomains Y1a and Y1b as one domain. Both subdomains form a unit in multimer prediction, which gives both subdomains a common function in the composition of the pore complex (see below). Furthermore, conservation of residues of Y1a + Y1b is higher than for the next domain, CoV-Y. The same is true for similarity for predicted structures (Table 3). CoV-Y is also tightly packed, but pAE indicates a clear divide between Y1 and CoV-Y at Gly1763 (see Fig 7B). Although Y1a + Y1b and CoV-Ya + CoV-Yb are packed tightly, Y1 and CoV-Y are connected via a potentially flexible hinge, which is also observable in the PDB structure 8F2E. The fold predictions are also supported by experimental evidence, with an RMSD of 0.4 Å between PDB entry 8F2E and each of the domains Y1b, CoV-Ya, and CoV-Yb, respectively.

For subdomain Y1a, no experimental structure has been solved so far, and our construct comprising Y1a + Y1b was toxic to *E. coli* during expression (data not included) so that we also were not able to measure it. This is unfortunate, as sequence analysis reveals a high degree of conservation among betacoronaviruses. Most of the conserved residues are in the loop region (residues 1,620–1,649), which is probably stable because of sidechain interactions. This

makes the fold vulnerable to mutations—meaning it must be highly conserved to maintain its fold. Moreover, four of the conserved residues form a cysteine–histidine cluster (three cysteines, one histidine), which is a potential zinc-binding site and could stabilize the fold (35). However, this binding site must be validated experimentally, as it may has been predicted falsely because of the limitations of AlphaFold2 regarding sidechain conformation prediction.

### Multimer prediction of Y1 and Y1+CoV-Y

Nsp3 assembles with nsp4 and nsp6 into a hexameric complex, which exports RNA from the replication organelle's interior into the cytosol (7). However, the question of what mediates the hexameric assembly of this biologically essential pore remains. We think it may be mediated by the Y1 domain, which becomes clear if considering the spatial arrangement of nsp3. It has two transmembrane domains and both N and C terminus are cytosolic. As Y1 is the first domain on the cytosolic side after the transmembrane region, and because no large linker precedes it, we can assume that this domain forms the foundation of the pore complex right on the membrane. For reasons discussed in the previous section, the domains before the transmembrane region cannot be located adjacent to the pore, leaving Y1 as the only option.

Our predicted model of Y1+CoV-Y (of which each subdomain agrees well with experimental structures) fits the diameter of the base of the pore (Fig 8). Because the electron tomographic structure cannot accommodate much more domains at the pore's base, the N-terminal cytosolic part of nsp3 must go elsewhere, but where exactly? As described in the previous sections, each of the two large linkers preceding the transmembrane region ($\beta$SM-N and $\beta$SM-C) would be long enough to locate the remaining cytosolic domains on top of the predicted Y1 structure.

Locating Y1+CoV-Y at the base would mean that these domains would need to form a hexameric ring with a channel, and in fact, multimer prediction leads to a high-confidence hexameric Y1 complex regarding pLDDT and pAE values. Furthermore, the diameter of the pore's channel of 2.2 nm (Fig 7E) agrees with the estimates of 2 to 3 nm based on the experimental evidence (7). The monomers are connected through hydrogen bonds between 11 residues in each monomer, of which three residues are conserved among all examined betacoronaviruses and a fourth remains similar in properties (40). The CoV-Y domains extend away from the channel and are not interacting with each other, leaving Y1 as the only C-terminal domain for holding the assembly together. This is supported by experimental evidence by Li et al, where a construct comprising Y1b + CoV-Y (PDB 8F2E) crystallized in the monomeric form (19).

Electrostatics of the predicted Y1 hexamer show a positively charged interior of the pore, which would make it suitable as RNA export channel, as RNA is negatively charged. The bottom surface of the hexamer shows a consistently positive charge, which would allow it to interact with the slightly negatively charged membrane of the replication organelle. Also, the hexamer fits well into the complex's base of the electron tomographic map (Fig 8), where membrane contact is unavoidable. Conclusively, we postulate that Y1 forms the foundation of the hexameric pore complex, whereas the remaining domains are assembling on top of Y1 and CoV-Y, with Y1 being the major contributor in the assembly.

### Experimental validation

Unfortunately, we were not able to purify Y1 in sufficient quantity to set up crystallization trials. Because of the little published information regarding the purification of the C-terminal domains, and the complete absence of an experimentally determined Y1 structure, we suspect that other researchers were also unsuccessful with this domain. Part of the problem could be the large positively charged surface emerging upon hexamer formation, which could bind to lipid membranes, as well as its potential ability to bind RNA and DNA.

Therefore, we suggest creating mutational variants of Y1 incapable of hexamer formation. Mutating contact residues to alanine might be insufficient, as AlphaFold2 still predicts hexamers with hydrogen bonds to backbone atoms. Mutating to proline, however, works at least in the predictions. Whether effects from mutations are predicted in AlphaFold2 correctly (26 *Preprint*, 41 *Preprint*) or not (42, 43) is still to debate. In any case, Y1 mutants with intact fold incapable of hexamer formation can be used for structure solution and for analysis of the hexamer's impact on the viral fitness, which can finally clarify Y1's status of a potential drug target.

### Conservation of Y1 among nidoviruses

Literature labels the Y1 domain as "nidovirus-conserved domain of unknown function." Furthermore, double-membrane vesicles as replication organelles appear in all positive-stranded RNA viruses (44), making the need for an RNA exporter channel present in a large variety of pathogens. Nsp3 has been shown to form an RNA exporter pore in MHV (7), suggesting that nsp3 equivalents in other coronaviruses can similarly form exporter pore complexes. Conservation in *Nidovirales* supports our previously defined hypothesis of Y1 forming the base of the hexameric RNA exporter channel. In the high-confidence prediction of the Y1 hexamer, multiple hydrogen bonds are predicted, which hold the monomers together. Many of these key residues, as well as an unusually high fraction of all Y1 residues, are conserved at least among *Orthocoronavirinae*. However, no similar structure predictions could be generated for viruses outside of *Orthocoronavirinae* and sequence similarities do not rise above 30%, opening the possibility of Y1 being not nidovirus-conserved. In such a case, related proteins with different folds, as well as alternative mechanisms for pore assembly, are thinkable.

Conclusively, our findings suggest Y1 to be a coronavirus-conserved domain likely involved in pore complex assembly. Because of its high conservation in *Orthocoronavirinae*, it is an interesting drug target not only in human-infecting betacoronaviruses, but also in gammacoronaviruses, which infect various bird species and are therefore a potential threat for livestock. Researching this domain could thus lead to therapeutics targeting a wide range of viral diseases.

### Final words

Nonstructural protein 3 is a large multidomain protein from SARS-CoV-2 with two established drug target domains and several domains with unresolved functions or structure. Its long evolutionary history and presence across distantly related viruses point at its vital function to the infection cycle, of which we may have uncovered only a fraction. Our discovery of the betacoronavirus-

specific linker domain underlines this aspect and demonstrates the need for better domain annotation. Information about the exploration status, such as "unexplored," "predicted," or "experimentally validated," would improve our understanding of multidomain proteins and highlight knowledge gaps. We demonstrated that automated use of modern structure prediction is an excellent starting point for domain annotation in large multidomain proteins and can also provide constructs for experimental validation.

Electron tomography of nsp3 shows that it forms a large hexameric pore for the export of RNA into the cytosol. Based on high-confidence predictions, we put forward the hypothesis that the Y1 domain drives the hexameric assembly. This is supported by conservation of key residues regarding assembly and folding, electrostatics, fitting the multimeric model into the experimental electron tomography map, and geometrical consideration regarding domain vicinity in the assembled complex. If purification of Y1 is successful, validation of the hexameric assembly by size exclusion is a logical next step. Evaluation of Y1's impact on the total number of pore complexes via mutation experiments will determine Y1's influence on viral fitness and clear its status as a potential drug target.

Research into the interaction between Y1 and other domains, such as NAB, βSM-M, and the betacoronavirus-specific linker domain, which could bind to the top part of Y1+CoV-Y, as well as the binding of Y1 hexamer to lipid membranes, may shed more light on nsp3 and the biological mechanisms of coronaviruses.

## Materials and Methods

### Using AlphaFold2 for domain boundary determination and construct design

We used the free access version of AlphaFold2 via Google Colab, ColabFold (37 Preprint), with default settings (without templates and without relaxation), using the MMSeqs2 algorithm (45) for multiple sequence alignment. For both sarbecoviruses, we used the version v2.1.1, whereas for MHV, we used v2.1.2. Toward the end of our work, v2.3.1 was released, which was used for submission of full-length nsp3 sequences because of a larger input sequence limit.

Protein sequences of SARS-CoV-2, SARS-CoV-1, and MHV nsp3 were obtained from NCBI (reference ids YP_009742610.1, NP_828862.2, and NC_048217.1, respectively).

The preliminary domain ranges for all nsp3 domains of SARS-CoV-1 were defined first, because it had the most experimentally determined domain structures in the PDB (46). The solved domains included Ubl1, Mac1, Mac2, Mac3, DPUP, Ubl2, PL2pro, and NAB. For prediction of transmembrane domains, the full nsp3 sequence was submitted to the TMHMM 2.0 server (30). Domain ranges of disordered domains or unresolved domains, that is, proposed domains not associated with experimentally determined structures, were taken from literature (4). The remaining regions between all domains were designated as linker regions.

The preliminary domain ranges of SARS-CoV-2 and MHV were defined by an analogous procedure with the respective domain structures from the PDB, the gene annotations in the NCBI entry, transmembrane domain prediction, and literature research. For SARS-CoV-2, the PDB (46) contained at the time of this work structures for the domains Ubl1, Mac1, Ubl2, PL2pro, NAB, part of βSM, and CoV-Yb. At the end of our work, the domains Y1b and complete CoV-Y were published, but not used for our initial domain determination. For MHV, PDB contained Ubl1, the DPUP-like domain, Ubl2, and PL2pro. To define the ranges of the remaining domains in SARS-CoV-2 and MHV, we performed global sequence alignments with Clustal Omega (47) between the preliminary ranges of SARS-CoV-1 and the full nsp3 sequence of the other two viruses. Each residue of all three nsp3 sequences was assigned to a domain or a linker between domains, respectively.

The sequences of each preliminary domain range were submitted to ColabFold (37 Preprint). To evaluate the prediction's accuracy, we aligned the models predicted by AlphaFold2 with the respective experimentally determined structures from the PDB (46) in PyMOL (31) and calculated there the root mean square deviation (Table 1).

The AlphaFold2 results and their pLDDT values were used to determine regions of order and disorder for all submitted sequences. The classification process is outlined in Fig 2. Regions with a lack of secondary structure elements and average pLDDT values below 50 were considered disordered, where secondary structure elements such as α-helices and β-sheets were recognized manually from cartoon representation in PyMOL (31), which were recognized by PyMOL's internal DSSP algorithm. These included also so-called "barbed-wire" regions with no proper torsion angles or hydrogen bonds as described by reference 28. Secondary structure elements with pLDDT above 80 and loops with pLDDT 50–80 flanked by secondary structure elements were considered ordered.

Residues in secondary structure elements (α-helices and β-sheets) with pLDDT values below 80 were evaluated individually. First, we truncated the potentially disordered regions at the termini and resubmitted the shortened sequence, which resulted in different pLDDT values. If the pLDDT of a region increased drastically (values rising from below 50 to over 80), the region was considered ordered henceforth. Regions containing α-helices where pLDDT stayed below 80 after cropping, and which pointed away from the main fold, were classified as potentially disordered, especially if the pLDDT lowered after cleavage. Regions containing β-sheets were classified as ordered.

Structure predictions containing large proportions of low pLDDT (such as βSM domain) were iteratively truncated and resubmitted, resulting in increasing pLDDT values for the folded regions. Iterative truncation of all domains was performed at the termini, if more than three terminal residues showed pLDDT values below 50. At each iteration, we cleaved regions we classified as disordered (via the method above) completely except for three residues at the border where the pLDDT rises above 50.

In addition to the individual domain ranges, we submitted sequences to AlphaFold2, which covered multiple domains and linkers (see Fig S1). These predictions contained two or more consecutive domains positioned relative to each other, and the

predicted alignment error (pAE) was used to assess the validity of such assemblies. Consistent values close to zero in the pAE matrix across all residues were considered as plausible alignment between two domains. Average pAE values above 15 were considered as implausible. Such alignment information was used to identify subdomains, which are part of a larger domain. To calculate average pLDDT and pAE values, we used a custom python script. All results together were considered for defining the final ranges of each individual domain, where the pAE was used last.

Ramachandran outliers for the predicted folds of the betacoronavirus-specific linker domain and the Y1 hexamer were calculated by MolProbity (33) to evaluate the prediction's validity.

### Experimental validation of the betacoronavirus-specific linker domain (βSLD)

Using the method described before, we identified a new folded domain between PL2^pro and NAB (residues 1,057–1,090 in SARS-CoV-2 nsp3), which was analyzed further with bioinformatics and experimental methods.

#### *Analysis of predicted structure and conservation of its sequence*
The new fold was checked visually in PyMOL (31), with a focus on hydrogen bonds by applying the preset "technical." The sequence of the new domain (residues 1,057–1,090 in SARS-CoV-2 nsp3) was submitted to BLAST (48), with SARS-CoV-2 sequences excluded. In additional BLAST queries, the whole taxonomy of *Coronaviridae* was excluded to find other homologs. Furthermore, we used the sequences of all betacoronaviruses found under the NCBI taxonomy id 694002 in local pairwise sequence alignments against the sequence of the new domain from SARS-CoV-2. The alignment was performed via EMBOSS Water version 6.6.0 (47). Target sequences from other betacoronaviruses were annotated equivalents of SARS-CoV-2 nsp3, if available. Otherwise, the sequence of the respective polyprotein 1ab was used, which contains all nsps. In the latter case, we ensured that the alignment was potentially localized within the nsp3 region on the polyprotein. The sequences from the alignment results were extended to match the length of the domain in SARS-CoV-2 and were afterward submitted to ColabFold. The whole procedure was repeated for the remaining *Coronaviridae*.

Because the two domains following the linker domain, NAB and βSM domain, were previously assumed to be betacoronavirus-specific based on sequence information alone (17), we applied the same approach on those two domains as well.

#### *Expression of nsp3 PL2^pro + LinkerDomain*
To validate the presence of the discovered linker domain, in vitro experiments were performed. The construct containing the domain Ubl2, PL2^pro, and the linker domain (SARS-CoV-2, Wuhan original strain; C111S mutant; see Supplemental Data 1 Section B) was amplified by PCR from a template kindly provided by David LV Bauer (Francis Crick Institute). The coding sequence was ligated into a pGEX-6P vector using (5′) BamHI and NotI (3′) restriction sites introduced during amplification. Correct insertion was confirmed by Sanger sequencing. The recombinant protein was expressed in LB in BL21Gold *E. coli* as a GST-3C-fusion protein. After

induction at $OD_{600}$ of 0.6, the temperature was reduced to 16°C, and cells were harvested the next morning. Cell pellets were frozen at –80°C until needed.

#### *Purification of nsp3 PL2^pro + LinkerDomain*
Cell pellets were resuspended in lysis buffer (50 mM Tris, pH 7.5, 300 mM NaCl, 5% [vol/vol] glycerol, 0.5 mM TCEP, 1 μM zinc acetate). EDTA-free protease inhibitors (Roche) were added as per manufacturer's instructions. Cells were lysed by sonication, and lysate was clarified by centrifugation at 45,000g, 4°C, for 45 min. The protein was harvested by incubating the lysate with glutathione Sepharose 4B (Cytiva) for 2 h at 4°C with constant mixing. The beads were then harvested and washed with 10 bed volumes of lysis buffer and then 20 bed volumes of SEC buffer (20 mM Tris, pH 7.5, 150 mM NaCl, 0.5 mM TCEP, 1 μM zinc acetate). Beads were then resuspended in five bed-volumes of wash buffer, and then incubated with GST-HRV3C protease overnight at 4°C with constant mixing. The cleaved product was collected from the beads via filtration. This material was then concentrated to 5 mg/ml, aliquoted, snap-frozen in liquid nitrogen, and stored at –80°C until needed.

#### *Crystallization of nsp3 PL2^pro + LinkerDomain*
Before crystallization, the protein was thawed on ice and any remaining aggregates or impurities were removed via a final size-exclusion chromatography step, using a Superdex 200 Increase column, equilibrated in SEC buffer (20 mM Tris–HCl, pH 7.5, 150 mM NaCl, 1 mM TCEP, 1 μM zinc acetate). This material was diluted 1:2 with Milli-Q water and concentrated to 7.5 mg/ml for crystallization trials. Sitting-drop vapor diffusion crystallization experiments were set up in MRC two-well 96-well plates using a Formulatrix NT-8 drop setting robot. Initial microcrystals were obtained using small (200 nl protein + 100 mother liquor) drops, which were then used to streak seed into larger 400 + 200 nl drops. Crystals appeared after ~4–5 wk after incubation and streak seeding at 4°C in 0.1 M MES, pH 6.8, 8.4% PEG20K. Crystals were cryocooled in liquid nitrogen using crystallization liquor supplemented with 20% ethylene glycol.

Diffraction data were collected at beamline I24 at Diamond Light Source. Autoprocessing using the autoPROC pipeline (49) indicated that the data extended to ~2.1 Å resolution, and that the crystals had space group P1. Molecular replacement using the MrBUMP pipeline (50) in CCP4 Cloud (51) gave a reasonable solution using PDB 4M0W ("Crystal Structure of SARS-CoV papain-like protease C112S mutant in complex with ubiquitin" (52)) with a TFZ of 12.0 and an initial $R_{free}$ of 47%.

Inspection of the electron density map in COOT (53) revealed that the amino-terminal lobe of PL2^pro was well defined, but the carboxy-terminal lobe was not, with poor electron density and noisy difference maps. The output from this molecular replacement solution was submitted to the ModelCraft auto-building and refinement pipeline (54), which finished with a final $R_{free}$ of 26.5%. Inspection of the results revealed that the crystals in fact contained two copies of the N-terminal lobe (Arg3 to Cys181 [construct numbering, Supplemental Data 1 Section B]) of the PL2^pro monomer, forming a close and compact dimer. No further refinement was undertaken.

### SAXS analysis of nsp3 PL2$^{pro}$ + LinkerDomain

The purified protein was shipped on dry ice to P12 BioSAXS beamline (55) at PETRA III (DESY), where the small-angle X-ray scattering (SAXS) experiment was conducted. P12 provides monochromatic X rays with a 0.2 × 0.05 mm$^2$ beam at the sample capillary. For this experiment, the X-ray wavelength and sample-to-detector distance were 0.124 and 3,000 mm, respectively. The purified monodisperse fraction was prepared as protein stocks with a concentration of 15 mg/ml. The stock buffer is 20 mM Tris (pH 8.5), 150 mM NaCl, 0.5 mM TCEP, 1 $\mu$M zinc acetate. The buffer for background subtraction was prepared by first diluting the protein stock 1:1 (vol:vol) with a close match of the stock buffer, then collecting the flow-through of ultrafiltration when the diluted protein solution was concentrated to the original volume. The concentration series (0.65 mg/ml, 0.97 mg/ml, 1.29 mg/ml, 1.61 mg/ml, 1.94 mg/ml, 3.22 mg/ml) is prepared by dilution with the closely matched buffer and careful filtration with Nanosep 100K OMEGA filters (PALL Life Sciences). The concentration series was measured under the standard "batch mode" at P12. The scattering profile was radially integrated, averaged, and absolutely scaled from the corresponding detector images of 30 exposures. Each exposure was 95 ms. The model and its theoretical SAXS curve are from the most fitted output generated by SREFLEX (34), using the background-subtracted SAXS profile of the 3.22 mg/ml sample and relaxed AlphaFold2 prediction as the input. The ab initio modeling was carried out as follows: (1) generate 20 bead models with DAMMIF (56) using the GNOM (57) output of the 3.22 mg/ml sample; (2) generate a starting search volume by averaging the 20 bead models using DAMAVER (58); (3) run DAMMIN (59) using the starting search volume and the GNOM output of the 3.22 mg/ml sample.

### Insights into the predicted hexameric assembly of Y1+CoV-Y

### Conservation of Y1 among Nidovirales

After determining new domain ranges for the Y1 domain, the nidovirus-conserved domain of unknown function, a protein–protein BLAST search (48) with Y1 from SARS-CoV-2, excluding SARS-CoV-2, was performed. Afterward, global sequence alignments with this sequence and the polyprotein sequences of other viruses were performed with Clustal Omega on default settings (47) to identify sequences of Y1 equivalents. These sequences were then submitted to AlphaFold2 (22) via ColabFold (37 Preprint). The predicted structures were aligned in PyMOL (31) to the structure prediction of Y1 from SARS-CoV-2 in order to calculate the root mean square deviation (RMSD) to measure the predicted structural similarity. The examined viruses are listed in Supplemental Data 1 Section B.

### Multimer prediction of Y1+CoV-Y

Based on the hexameric assembly of nsp3 in complex with nsp4 and nsp6, which was observed by cryoelectron tomography (7), and Y1's potential location at the pore's base, a hexamer prediction of the Y1 domain was performed via AlphaFold2's multimer feature (36 Preprint). It was afterward repeated for each other domain. Because the prediction did not complete for large domains at default settings, the code in Colab was adapted to output only one model by setting "num_models" to 1 and model_

order to "(1)." The confidence of multimeric arrangements was evaluated with the pAE matrix, where average pAE values below 10 were handled as confident. Electrostatics of confidently predicted hexamers were calculated in PyMOL (31) via "generate vacuum electrostatics."

Fitting the models to the cryoelectron tomography density map was accomplished in UCSF Chimera (38), by first positioning it manually at the base of the pore complex in the correct orientation and afterward applying Chimera's fitting algorithm under Tools, Volume Data, Fit in Map. Rotation and shift were allowed.

### Expression and purification of Y1

The construct of Y1 (Supplemental Data 1 Section C) was amplified by PCR from a template kindly provided by David LV Bauer (Francis Crick Institute). The coding sequence was ligated into a pGEX-6P vector using (5′) BamHI and NotI (3′) restriction sites introduced during amplification. Correct insertion was confirmed by Sanger sequencing. Recombinant protein was expressed in LB in BL21Gold E. coli as a GST-3C-fusion protein. After induction at OD$_{600}$ of 0.6, the temperature was reduced to 16°C, and cells were harvested the next morning. Cell pellets were frozen at –80°C until needed.

Cell pellets were resuspended in lysis buffer (50 mM Tris, pH 7.5, 300 mM NaCl, 5% [vol/vol] glycerol, 0.5 mM TCEP, 1 $\mu$M zinc acetate). EDTA-free protease inhibitors (Roche) were added as per manufacturer's instructions. Cells were lysed by sonication, and lysate was clarified by centrifugation at 45,000$g$, 4°C, for 45 min. The protein was harvested by incubating the lysate with glutathione Sepharose 4B (Cytiva) for 2 h at 4°C with constant mixing. The beads were then harvested and washed with 10 bed volumes of lysis buffer and then 20 bed volumes of SEC buffer (20 mM Tris, pH 7.5, 150 mM NaCl, 0.5 mM TCEP, 1 $\mu$M zinc acetate). Beads were then resuspended in five bed volumes of wash buffer, and then incubated with GST-HRV3C protease overnight at 4°C with constant mixing. Analysis by SDS–PAGE of the purification process showed that no detectable Y1 had been expressed.

## Data Availability

All data are available in the main text or Supplemental Data 1. Raw SAXS data, scripts, and other unique resources are available upon request to the authors.

## Supplementary Information

## Acknowledgements

Thanks to Gianluca Santoni, Lea von Soosten, the Coronavirus Structural Taskforce (20), Arwen Pearson, Andrew Torda, and Kay Grünewald for interesting and insightful discussions. Huge thanks goes to David LV Bauer from Francis Crick Institute, who provided us the template for our PL2$^{pro}$ +

LinkerDomain construct. We thank Diamond Light Source for beamtime (proposal mx25587) and the involved staff at I24 for assisting with data collection. We acknowledge EMBL Hamburg (DESY) for kindly providing emergency beamtime. We would like to give special thanks to Cy Jeffries (EMBL, Hamburg) for his assistance over the course of the BioSAXS experiments. The beamtime was allocated as a part of the SPC-CSSB BAG proposal. This work was supported by the German Federal Ministry of Education and Research (grant no. 05K19WWA and 05K22GU5) and Deutsche Forschungsgemeinschaft (project TH2135/2-1). DC Briggs acknowledges that this work was supported by the Francis Crick Institute, which receives its core funding from Cancer Research UK (CC2068), the UK Medical Research Council (CC2068), and the Wellcome Trust (CC2068).

## Author Contributions

M Edich: conceptualization, resources, data curation, software, formal analysis, validation, visualization, methodology, and writing—original draft, review, and editing.
Y Gao: conceptualization, resources, data curation, software, formal analysis, supervision, validation, investigation, visualization, methodology, and writing—review and editing.
DC Briggs: conceptualization, resources, data curation, software, formal analysis, supervision, validation, investigation, methodology, and writing—original draft, review, and editing.
A Thorn: conceptualization, resources, formal analysis, supervision, funding acquisition, validation, investigation, visualization, methodology, project administration, and writing—original draft, review, and editing.

## Conflict of Interest Statement

The authors declare that they have no conflict of interest.

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
