## [Reviewer comments · Life Science Alliance]

Unlocking the Secrets of SARS-CoV-2 nsp3 by Combining Experiments with AlphaFold2 Domain Prediction

Maximilian Edich, Yunyun Gao, David Briggs, and Andrea Thorn

DOI: <https://doi.org/10.26508/lsa.202503247>

Corresponding author(s): Andrea Thorn, Helmholtz-Zentrum Berlin für Materialien und Energie

Review Timeline:

Submission Date:	2025-02-04
Editorial Decision:	2025-03-21
Revision Received:	2025-05-08
Editorial Decision:	2025-06-16
Revision Received:	2026-01-09
Accepted:	2026-01-14

Scientific Editor: Tim Fessenden

Transaction Report:

March 21, 2025

Re: Life Science Alliance manuscript #LSA-2025-03247-T

Dr. Andrea Thorn
Institute for Nanostructure and Solid State Physics, University of Hamburg
Germany

Dear Dr. Thorn,

Thank you for submitting your manuscript entitled "Unlocking the Secrets of NSP3: AlphaFold2-assisted Domain Determination in SARS-CoV-2 Protein" to Life Science Alliance. The manuscript was assessed by expert reviewers, whose comments are appended to this letter. We invite you to submit a revised manuscript addressing the Reviewer comments.

Thank you for this interesting contribution to Life Science Alliance. We are looking forward to receiving your revised manuscript.

Sincerely,

B. MANUSCRIPT ORGANIZATION AND FORMATTING:

Reviewer #1 (Comments to the Authors (Required)):

The manuscript describes a massive effort to predict the structures of nsp3, the largest protein from SARS-CoV-2, by alphafold predictions. From these predictions, the authors identified a yet not characterization domain of nsp3. The provide evidence of the folded state of the protein by low resolution SAXS data and Xray data. Given the aspect that sufficient material could be obtained, it is a pity that the authors did not use NMR to characterize the protein structure.

As the work that is presented in this manuscript is a massive undertaking, the authors can be applauded to describe the (vast) amount of experimental data on nsp3 in a fair manner.

Citation to important work by NMR is a little underrepresented, in particular from the Blackledge lab. The authors might want to considered adding reference to this work during the galley stage of the processing of the manuscript.

Else, I support publication of this nice work, essentially as is.

Reviewer #2 (Comments to the Authors (Required)):

This is a useful study that combines AlphaFold2 structure predictions with available NMR and X-ray crystal structure data and phylogenetic data to predict domain boundaries in the large, enigmatic coronavirus protein nsp3.

Unlike some other primarily computational and predictive studies on this subject, this study has the benefit of experimental validation, in the form of a small angle X-ray scattering model of a domain that extends a previously-solved domain structure. Overall, I found it insightful and compellingly written.

My comments below are minor.

Intro - RNA being "Packed into nucleoproteins" doesn't make sense - interacts with? Ribonucleoprotein complexes form?

Conservation of Y1 - misspells Neuman. Ask me how I know.

Fig. 1 - I don't know if DMV lumen is the right term here, since 3Ecto would sit between the two membranes of the DMV. Inter-membrane space? You may be able to come up with something more graceful.

The original SARS-CoV is still just called SARS-CoV, not SARS-CoV-1. The potential for confusion was discussed at the time SARS-CoV-2 was named, but that's just what the committee decided to do.

Fig. 3 - the change in scale between panels is a little jarring - could you show these at the same scale please?

For what it's worth, Serrano et al 2007 did show the hypervariable region is flexibly disordered, and their NMR data for the N-terminal region of nsp3 before the Ub1 globular domain is a good match for your predictions. The structures of both are, in essence, solved. The N-terminal loop is discussed adequately, but the HVR kind of gets left out. I don't know that it warrants a change in Fig. 1, but maybe a mention in the text.

If I am correct in thinking AlphaFold is not (yet) able to account for structurally significant heterogens like zinc, the problem with the N-terminal boundary of Y1a is not so much sequence ambiguity as (if I understand correctly) AlphaFold's inherent limitation. Universally conserved, possibly novel, non-RING zinc fingers would be a pretty big blind spot for AlphaFold, and worth discussing as a caveat for this section of the prediction.

MHV is a pretty diverse collection of viruses with at least some minor variation in nsp3 - best to specify a strain. Neither SARS-CoV (assuming we are talking about the human-associated strains) nor SARS-CoV-2 are diverse enough to need any further specification.

Is there a reason to split Torovirinae and Piscanivirinae off from the Tobaniviridae? I think (though I would need to check) that any conservation would also apply to the Serpentovirinae and Remotovirinae, and then you can just mention the family. Or if there is a distinct lack of homology apparent in the other two subfamilies, that would be worth noting.

Fig. 6C - I'm not so familiar with SAXS structures, so this question may not make much sense. How did you set the isosurface threshold for the grey SAXS model in 6C? It seems to have an enormous volume compared to the fitted protein structure. How much of that difference is accounted for in sidechain volumes that are not rendered on the fitted model? I think you are right about the similarity, but since the similarity is the main experimental validation in this study, it might be nice to have it look more convincing with the sidechains in there.

Fig. 8 - I think your rotation arrows may be backwards - unless I'm suffering from some kind of optical illusion, the model seems to be tilting down rather than up in 45 degree increments.

Discussion - calling your models "cheap" seems a little more self-deprecating than it needs to be. Computationally inexpensive, rapid, predictive, hypothesis-generating - seems like there are other ways you could go with the wording.

Reviewer #3 (Comments to the Authors (Required)):

The manuscript presents a comprehensive study of non-structural protein 3 (nsp3) in SARS-CoV-2 and its homologs, combining analysis of sequence data and structural information from experimental methods and computational predictions (AlphaFold2, ColabFold). The integration of AI-based tools with experimental data allows to fill some of the knowledge gaps in the nsp3 structure and function. The work offers more refined definitions of nsp3 domains, identifies new potentially structured elements in the protein (betacoronavirus-specific linker domain) and proposes a role of the Y1 domain in nsp3 hexamerization based on the structural predictions. The authors attempted to validate two of their findings experimentally (determine the structure of β SLD and Y1 hexamer) but, with only low resolution SAXS data for Ubl2-PL2pro- β SLD and inability to obtain the Y1 sample, this has not been entirely successful. Nevertheless, the study offers a useful starting point for the follow up research.

I'd recommend some improvements as listed below:

1). The authors incorrectly use the term " β -sheet" - in most cases (not all) they seem to mean either a β -strand or a β -hairpin.

For example, in section "Experimental validation of the Betacoronavirus-specific linker domain (β SLD)" in Results, the sentence:

"Additionally, AlphaFold2 predicts a stable fold consisting of 34 residues for SARS-CoV-2, forming two pairs of antiparallel β -sheets connected by an 8-residue loop (Figure 4)"

Should read:

"Additionally, AlphaFold2 predicts a stable fold consisting of 34 residues for SARS-CoV-2, forming two β -hairpins connected by an 8-residue loop (Figure 4)"

Farther in the same section, the below text is not needed if the β 1- β 2 and β 3- β 4 hairpins are properly identified elsewhere (by the β -hairpin definition they must be connected by the backbone hydrogen bonds)

"The sheet β 1 is connected to β 2 via hydrogen bonds in the backbone and..."

The description should be cross-checked with the "Experimental validation of the Betacoronavirus-specific linker domain (β SLD)" section of Discussion.

The below section in "Structure prediction of Y1 and CoV-Y" also needs to be revised:

"The fold of Y1a comprises two large β -sheets (21 residues in length), a 50-residues region of loops, and an 11-residue long α -helix (Fig. 7d). This globular fold consisting mostly of loops is held together by numerous hydrogen bonds and contains a conserved cysteine-histidine cluster, which could stabilize the fold by binding metal ions (Fig. 7c).

Y1b makes up the upper half of Y1 (90 residues), containing a triplet of parallel β -sheets, which creates an intertwined structure with a triplet of anti-parallel β -sheets, an α -helix, a loop stabilized by H-bonds, and sharp turns.

The subdomain CoV-Ya consists of four α -helices and CoV-Yb of four α -helices, four β -sheets, and various loops and turns, resulting in a globular fold (Fig. 7a)."

The "two large β -sheets (21 residues in length)" seem to be a β -hairpin (if I judge properly from the figure). The Y1b subdomain simply contains two β -sheets - one parallel and one antiparallel, each consisting of 3 β -strands. CoV-Yb has 6 β -strands packed into two β -sheets, one antiparallel and one mixed.

The above are only examples, the entire text should be revisited to ensure proper terminology for structural elements.

2). Fig. 1 - I think the content of a and b panel could be combined as there are duplications in the content. Also, the coloring scheme for the predicted structures meant to show confidence is not visible, so perhaps could be dropped.

3). Please, address minor editing:

- use either "nsp3" or "NSP3" to name the protein, currently both variants are present
- same with AlphaFold2 vs Alphafold2, Figure vs Fig. in the text
- The sentence "While the terminal linkers are likely disordered due to low pLDDT" should be revised as low pLDDT is only an indication of the disorder, not the reason for it
- "Y1a+Y1b was toxic to *E. coli* during" - please, use italic for *E. coli*
- In methods - use consistent spacing (50 mM vs 50mM), zinc acetate not Zinc Acetate, also check symbols for microliters
- It might be rather a request to journal than to the authors, but better differentiation in the major sections and subsections would be helpful. For example, "Nucleic acid binding domain and Betacoronavirus-specific marker domain" seems to be a subsection of "Experimental validation of the Betacoronavirus-specific linker domain (β SLD)", which I don't think is the case.

Reviewer #1 (Comments to the Authors (Required)):

The manuscript describes a massive effort to predict the structures of nsp3, the largest protein from SARS-CoV-2, by alphafold predictions. From these predictions, the authors identified a yet not characterization domain of nsp3. They provide evidence of the folded state of the protein by low resolution SAXS data and Xray data. Given the aspect that sufficient material could be obtained, it is a pity that the authors did not use NMR to characterize the protein structure.

As the work that is presented in this manuscript is a massive undertaking, the authors can be applauded to describe the (vast) amount of experimental data on nsp3 in a fair manner.

Citation to important work by NMR is a little underrepresented, in particular from the Blackledge lab. The authors might want to consider adding reference to this work during the galley stage of the processing of the manuscript.

Else, I support publication of this nice work, essentially as is.

Answer: Thank you very much for the review and the comments! We have added NMR to the section Non-structural-protein 3 domains and added the following references::

¹H, ¹³C and ¹⁵N backbone chemical shift assignments of SARS-CoV-2 nsp3a. (*Salvi et al., 2021*)

The intrinsically disordered SARS-CoV-2 nucleoprotein in dynamic complex with its viral partner nsp3a (*Bessa et al., 2022*).

NMR Provides Unique Insight into the Functional Dynamics and Interactions of Intrinsically Disordered Proteins (*Camacho-Zarco et al., 2022*).

Reviewer #2 (Comments to the Authors (Required)):

This is a useful study that combines AlphaFold2 structure predictions with available NMR and X-ray crystal structure data and phylogenetic data to predict domain boundaries in the large, enigmatic coronavirus protein nsp3.

Unlike some other primarily computational and predictive studies on this subject, this study has the benefit of experimental validation, in the form of a small angle X-ray scattering model of a domain that extends a previously-solved domain structure. Overall, I found it insightful and compellingly written.

My comments below are minor.

Intro - RNA being "Packed into nucleoproteins" doesn't make sense - interacts with? Ribonucleoprotein complexes form?

Answer: Revised.

Conservation of Y1 - misspells Neuman. Ask me how I know.

Answer: Sorry, corrected. Did you really swab white-tail deer?

Fig. 1 - I don't know if DMV lumen is the right term here, since 3Ecto would sit between the two membranes of the DMV. Inter-membrane space? You may be able to come up with something more graceful.

Answer: We changed DMV lumen to DMV intermembrane space.

The original SARS-CoV is still just called SARS-CoV, not SARS-CoV-1. The potential for confusion was discussed at the time SARS-CoV-2 was named, but that's just what the committee decided to do.

Answer: We know, but we find SARS-CoV-1 useful for distinction.

Fig. 3 - the change in scale between panels is a little jarring - could you show these at the same scale please?

Answer: We have updated Fig. 3.

For what it's worth, Serrano et al 2007 did show the hypervariable region is flexibly disordered, and their NMR data for the N-terminal region of nsp3 before the Ub1 globular domain is a good match for your predictions. The structures of both are, in essence, solved. The N-terminal loop is discussed adequately, but the HVR kind of gets left out. I don't know that it warrants a change in Fig. 1, but maybe a mention in the text.

Answer: We have added a reference to the suggested work in the caption.

If I am correct in thinking Alphafold is not (yet) able to account for structurally significant heterogens like zinc, the problem with the N-terminal boundary of Y1a is not so much sequence ambiguity as (if I understand correctly) Alphafold's inherent limitation. Universally conserved, possibly novel, non-RING zinc fingers would be a pretty big blind spot for Alphafold, and worth discussing as a caveat for this section of the prediction.

Answer: We have added:

“However, this binding site must be validated experimentally, as it may have been predicted falsely due to the limitations of AlphaFold2 regarding side-chain conformation prediction.”

after the corresponding discussion.

8. MHV is a pretty diverse collection of viruses with at least some minor variation in nsp3 - best to specify a strain. Neither SARS-CoV (assuming we are talking about the human-associated strains) nor SARS-CoV-2 are diverse enough to need any further specification.

Answer: We have specified the strain as follows “murine hepatitis virus strain A59 (MHV)”

9. Is there a reason to split Torovirinae and Piscanivirinae off from the Tobaniviridae? I think (though I would need to check) that any conservation would also apply to the Serpentovirinae and Remotovirinae, and then you can just mention the family. Or if there is a distinct lack of homology apparent in the other two subfamilies, that would be worth noting.

Answer: We thank reviewer's comment. We revised “Torovirinae ... Piscanivirinae” with “...Tobaniviridae”

10. Fig. 6C - I'm not so familiar with SAXS structures, so this question may not make much sense. How did you set the isosurface threshold for the grey SAXS model in 6C? It seems to have an enormous volume compared to the fitted protein structure. How much of that difference is accounted for in sidechain volumes that are not rendered on the fitted model? I think you are right about the similarity, but since the similarity is the main experimental validation in this study, it might be nice to have it look more convincing with the sidechains in there.

Answer: The SAXS envelope took the hydration shell of proteins into consideration. It is normal that the crystal structures (especially for the cartoon representation) appear “too small” than their SAXS envelopes. In fact, the fitting would be lousy if the hydration layer would not be considered. Besides this, under our experimental settings, the SAXS profile records a sum of protein dynamics, hence the SAXS envelope represents a total conformational space the protein can explore over the exposure period. We have also shown in the Kratky plot that, in solution, the protein has a certain flexibility.

However, the Kratky plot would look very differently if the linker domain would not be stably folded.

There is another cosmetic reason too: plotting with side chain simply makes the whole rendering ugly.

Fig. 8 - I think your rotation arrows may be backwards - unless I'm suffering from some kind of optical illusion, the model seems to be tilting down rather than up in 45 degree increments.

Answer: We confirm that the rotation indicator in the manuscript was correctly drawn as intended. To better guide readers, we have changed the arrows, added a perspective description and updated the caption accordingly.

12. Discussion - calling your models "cheap" seems a little more self-deprecating than it needs to be. Computationally inexpensive, rapid, predictive, hypothesis-generating - seems like there are other ways you could go with the wording.

Answer: You're right! We exchanged "cheap" with "hypothesis-generating."

Thank you very much for your comments and suggestions, they were great!

Reviewer #3 (Comments to the Authors (Required)):

The manuscript presents a comprehensive study of non-structural protein 3 (nsp3) in SARS-CoV-2 and its homologs, combining analysis of sequence data and structural information from experimental methods and computational predictions (AlphaFold2, ColabFold). The integration of AI-based tools with experimental data allows to fill some of the knowledge gaps in the nsp3 structure and function. The work offers more refined definitions of nsp3 domains, identifies new potentially structured elements in the protein (betacoronavirus-specific linker domain) and proposes a role of the Y1 domain in nsp3 hexamerization based on the structural predictions. The authors attempted to validate two of their findings experimentally (determine the structure of β SLD and Y1 hexamer) but, with only low resolution SAXS data for Ubl2-PL2pro- β SLD and inability to obtain the Y1 sample, this has not been entirely successful. Nevertheless, the study offers a useful starting point for the follow up research.

I'd recommend some improvements as listed below:

1). The authors incorrectly use the term " β -sheet" - in most cases (not all) they seem to mean either a β -strand or a β -hairpin.

For example, in section "Experimental validation of the Betacoronavirus-specific linker domain (β SLD)" in Results, the sentence:

Answer: Revised as suggested.

"Additionally, AlphaFold2 predicts a stable fold consisting of 34 residues for SARS-CoV-2, forming two pairs of antiparallel β -sheets connected by an 8-residue loop (Figure 4)"
Should read:

"Additionally, AlphaFold2 predicts a stable fold consisting of 34 residues for SARS-CoV-2, forming two β -hairpins connected by an 8-residue loop (Figure 4)"

Answer: Amended.

Farther in the same section, the below text is not needed if the β 1- β 2 and β 3- β 4 hairpins are properly identified elsewhere (by the β -hairpin definition they must be connected by the backbone hydrogen bonds)

"The sheet β 1 is connected to β 2 via hydrogen bonds in the backbone and..."

Answer: We thank the reviewer for the suggestion. Revised as suggested.

The description should be cross-check with the "Experimental validation of the Betacoronavirus-specific linker domain (β SLD)" section of Discussion.

Answer: Amended.

The below section in "Structure prediction of Y1 and CoV-Y" also needs to be revised:

"The fold of Y1a comprises two large β -sheets (21 residues in length), a 50-residues region of loops, and an 11-residue long α -helix (Fig. 7d). This globular fold consisting mostly of loops is held together by numerous hydrogen bonds and contains a conserved

cysteine-histidine cluster, which could stabilize the fold by binding metal ions (Fig. 7c). Y1b makes up the upper half of Y1 (90 residues), containing a triplet of parallel β -sheets, which creates an intertwined structure with a triplet of anti-parallel β -sheets, an α -helix, a loop stabilized by H-bonds, and sharp turns.

The subdomain CoV-Ya consists of four α -helices and CoV-Yb of four α -helices, four β -sheets, and various loops and turns, resulting in a globular fold (Fig. 7a)."

The "two large β -sheets (21 residues in length)" seem to be a β -hairpin (if I judge properly from the figure). The Y1b subdomain simply contains two β -sheets - one parallel and one antiparallel, each consisting of 3 β -strands. CoV-Yb has 6 β -strands packed into two β -sheets, one antiparallel and one mixed.

The above are only examples, the entire text should be revisited to ensure proper terminology for structural elements.

Answer: Thank you! We have revised the improper terminology, especially those on " β -sheets" and " β -strands", accordingly.

2). Fig. 1 - I think the content of a and b panel could be combined as there are duplications in the content. Also, the coloring scheme for the predicted structures meant to show confidence is not visible, so perhaps could be dropped.

Answer: We have updated Fig. 1 accordingly.

3). Please, address minor editing:

- use either "nsp3" or "NSP3" to name the protein, currently both variants are present

Answer: We have unified the nomenclature as "nsp3".

- same with AlphaFold2 vs Alphafold2, Figure vs Fig. in the text

Answer: We have unified the nomenclature to "AlphaFold2".

- The sentence "While the terminal linkers are likely disordered due to low pLDDT" should be revised as low pLDDT is only an indication of the disorder, not the reason for it

Answer: We have changed the sentence as follows:

"While the terminal linkers β SM-N and β SM-C are likely disordered due to low sequence identity in alignments, which is also reflected in low pLDDT values,..."

- "Y1a+Y1b was toxic to *E. coli* during" - please, use italic for *E. coli*

Answer: Amended.

- In methods - use consistent spacing (50 mM vs 50mM), zinc acetate not Zinc Acetate, also check symbols for microliters

Answer: Corresponding inconsistencies have been revised.

- It might be rather a request to journal than to the authors, but better differentiation in the major sections and subsections would be helpful. For example, "Nucleic acid binding domain and Betacoronavirus-specific marker domain" seems to be a subsection of "Experimental validation of the Betacoronavirus-specific linker domain (β SLD)", which I don't think is the case.

Answer: We would be glad to hear the editor's suggestion and amend accordingly!

Thank you very much for the review and the comments!

June 16, 2025

RE: Life Science Alliance Manuscript #LSA-2025-03247-TR

Dr. Andrea Thorn
Universität Hamburg
Institut für Nanostruktur und Festkörperphysik
Luruper Chaussee 149
Bldg. 610 (HARBOR)
Hamburg 22761
Germany

Dear Dr. Thorn,

Thank you for submitting your revised manuscript entitled "Unlocking the Secrets of nsp3: AlphaFold2-assisted Domain Determination in SARS-CoV-2 Protein" and for your patience during the re-review process. We would be happy to publish your paper in Life Science Alliance pending final revisions necessary to meet our formatting guidelines.

- Please be sure that the authorship listing and order is correct.
- Please add the X and Bluesky handles of your host institute/organization as well as your own or/and one of the authors in our system.
- Please remove track changes from the manuscript file and upload a clean version.
- It is recommended to exclude figures from the manuscript text and leave them uploaded separately only.
- Please add your main, supplementary figure, and table legends to the main manuscript text after the references section.
- Please consult our manuscript preparation guidelines <https://www.life-science-alliance.org/manuscript-prep> and make sure your manuscript sections are labeled correctly.
- Please incorporate any points from the Conclusion section into the Discussion; we only allow a Discussion section.
- Please add callouts for Figure 8D-G to your main manuscript text.
- Please add an Author Contributions section to your main manuscript text.
- Please add a Conflict of Interest statement to your main manuscript text.
- LSA does not permit colons in paper titles. Please amend your title accordingly, for instance "AlphaFold2-assisted Domain Determination in SARS-CoV-2 Nsp3" or similar.

A. FINAL FILES:

-- Summary blurb (enter in submission system): A short text summarizing in a single sentence the study (max. 200 characters including spaces). This text is used in conjunction with the titles of papers, hence should be informative and complementary to the title. It should describe the context and significance of the findings for a general readership; it should be written in the

present tense and refer to the work in the third person. Author names should not be mentioned.

B. MANUSCRIPT ORGANIZATION AND FORMATTING:

Sincerely,

Reviewer #2 (Comments to the Authors (Required)):

This revision answers the questions that I had - happy to go forward with publication. And to answer the authors' question, it was Francisco who swabbed the deer.

January 14, 2026

RE: Life Science Alliance Manuscript #LSA-2025-03247-TRR

Dr. Andrea Thorn
Helmholtz-Zentrum Berlin für Materialien und Energie
Department AI and Biomolecular Structures
Albert-Einstein Str. 15
Berlin 12489
Germany

Dear Dr. Thorn,

Thank you for submitting your Research Article entitled "Unlocking the Secrets of SARS-CoV-2 nsp3 by Combining Experiments with AlphaFold2 Domain Prediction". It is a pleasure to let you know that your manuscript is now accepted for publication in Life Science Alliance. Congratulations on this interesting work.

Your manuscript will now progress through copyediting and proofing. The convention at LSA is that section sub-headings are permitted only in the Results section. We therefore request that you omit the sub-headings in the Introduction and the Discussion. Please execute this change during the proofing process. We regret that this was not raised to you sooner, and we sincerely appreciate your understanding that this important work must still align with our journal's style conventions.

DISTRIBUTION OF MATERIALS:

Again, congratulations on a very nice paper. I hope you found the review process to be constructive and are pleased with how the manuscript was handled editorially. We look forward to future exciting submissions from your lab.

Sincerely,
